# Multi-omic features of oesophageal adenocarcinoma in patients treated with preoperative neoadjuvant therapy

Marjan M. Naeini [1], Felicity Newell [1], Lauren G. Aoude[2], Vanessa F. Bonazzi [2], Kalpana Patel[2], Guy Lampe[3], Lambros T. Koufariotis[1], Vanessa Lakis[1], Venkateswar Addala [1], Olga Kondrashova [1], Rebecca L. Johnston [1], Sowmya Sharma[1,4,5], Sandra Brosda [2], Oliver Holmes[1], Conrad Leonard [1], Scott Wood [1], Qinying Xu[1], Janine Thomas[3,6], Euan Walpole[3], G. Tao Mai[3], Stephen P. Ackland[7], Jarad Martin[8], Matthew Burge[9], Robert Finch[9], Christos S. Karapetis[10], Jenny Shannon[11], Louise Nott [12], Robert Bohmer[13], Kate Wilson[14], Elizabeth Barnes[14], John R. Zalcberg[15], B. Mark Smithers [3,4], John Simes[14], Timothy Price[16], Val Gebski [14], Katia Nones [1], David I. Watson [17], John V. Pearson[1], Andrew P. Barbour [2,3,18] ✉ & Nicola Waddell [1,18] ✉

Oesophageal adenocarcinoma is a poor prognosis cancer and the molecular features underpinning response to treatment remain unclear. We investigate whole genome, transcriptomic and methylation data from 115 oesophageal adenocarcinoma patients mostly from the DOCTOR phase II clinical trial (Australian New Zealand Clinical Trials Registry-ACTRN12609000665235), with exploratory analysis pre-specified in the study protocol of the trial. We report genomic features associated with poorer overall survival, such as the APOBEC mutational and RS3-like rearrangement signatures. We also show that positron emission tomography non-responders have more sub-clonal genomic copy number alterations. Transcriptomic analysis categorises patients into four immune clusters correlated with survival. The immune suppressed cluster is associated with worse survival, enriched with myeloid-derived cells, and an epithelial-mesenchymal transition signature. The immune hot cluster is associated with better survival, enriched with lymphocytes, myeloid-derived cells, and an immune signature including *CCL5*, *CD8A*, and *NKG7*. The immune clusters highlight patients who may respond to immunotherapy and thus may guide future clinical trials.

The incidence of oesophageal adenocarcinoma (OAC) is rising in Western countries, and OAC has one of the poorest long-term outcomes of all solid tumours[1]. Curative treatment based on oesophagectomy is only suitable for ~50% of patients due to aggressiveness and late-stage diagnosis. The addition of pre-operative chemotherapy or chemoradiotherapy has improved survival in OAC patients[2,3], and immune checkpoint blockade (ICB) therapy is showing some promise[4,5], however, it has yet to revolutionize treatment for OAC as in cancers such as melanoma and lung cancer[6,7]. Therefore, it is important to understand the molecular basis of treatment response

and risk-stratify patients to tailor therapy if there is to be an advance in treatment options.

Whole genome and exome sequencing studies have shown that OAC is a cancer with a high mutation burden and widespread chromosomal instability[8–11]. DNA methylation is a key component of epigenetic mechanisms for maintaining genome instability and regulating gene expression[12] and distinct methylation subgroups with prognostic implications have been described in OAC[13]. The genome instability in OAC includes frequent genomic catastrophes such as complex rearrangements, chromothripsis, breakage-fusion-bridge (BFB), and localized hypermutation termed kataegis[9,11]. Mutational signature analysis implicates multiple DNA repair processes involved in OAC tumorigenesis[11]. Signature 17 is a dominant signature in OAC[11], it can be an early event as was detected in Barrett's oesophagus[14,15], and can also occur later as has been associated with 5-fluorouracil (5-FU) treatment[16]. Other DNA repair mutation signatures reported in OAC include signatures associated with the AID/APOBEC, mismatch repair (MMR) deficiencies and homologous recombination repair deficiency (HRD) linked to *BRCA1* and/or *BRCA2* mutations. The studies to date help in explaining the tumourigenesis of OAC; however, an understanding of these complex genomic events in the context of treatment and patient outcome is still lacking.

In this work we study genomic, transcriptomic and methylation data from 115 OAC pre-treatment tumours, most of the patients were enrolled in the DOCTOR trial[17], from the Australasian Gastro-Intestinal Trials Group (AGITG). The DOCTOR trial was a randomized and non-comparative phase II clinical trial of docetaxel, cisplatin, and 5-FU (DCF) with or without radiotherapy (RT) based on poor early metabolic responses, which found the addition of docetaxel and RT improved the histological response rate, progression-free survival (PFS) and reduced recurrence in PET (positron emission tomography) non-responders. In this study, we analyse genomic, transcriptomic and methylation data with high-quality clinical information from the DOCTOR trial to identify prognostic biomarkers of treatment and stratify patients with potential therapeutic relevance. Furthermore, we identify OAC immune subtypes that correlate with patient outcomes and may be relevant for precision immunotherapy in the future.

## Results

We performed whole-genome sequencing (WGS), RNA sequencing (RNA-seq) and methylation profiling in the OAC tumour biopsies of 115 patients (Supplementary Data 1 and 2). WGS was performed on 89 patients, RNA-seq for 79 patients and methylation profiling for 72 patients. All patients received platinum-based chemotherapy as part of their treatment, with 92 patients enrolled in the DOCTOR clinical trial (Fig. 1a). Tumour biopsies were collected prior to treatment and surgery (Fig. 1b). The mean overall survival was 34.2 months (range 2–68 months) (Fig. 1b), and stage was significantly associated with overall survival (OS) (Fig. 1c). Clinical features of the cohort are described in Supplementary Data 1.

### Mutational landscape of oesophageal adenocarcinoma

The mean tumour single-nucleotide variant (SNV) and small insertion and deletion (indel) mutation burden for the samples was 8.35 mutations/Mb (range 1.03–41.34 mutations/Mb) (Fig. 2a), similar to previous reports in oesophageal cancer. Consistent with previous reports[18], we found a significant correlation between the number of somatic SNVs and the predicted number of neoantigens (Pearson

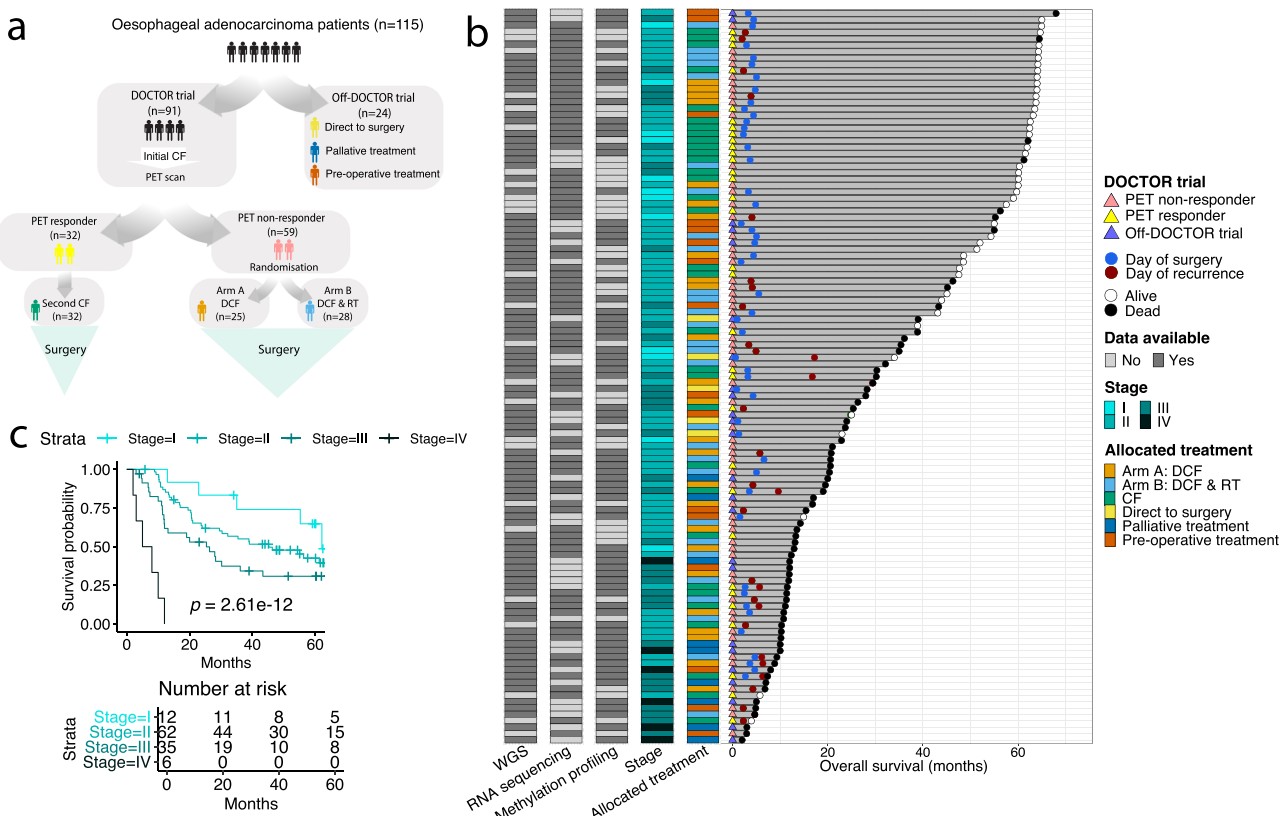

**Fig. 1 | Clinical overview of OAC cohort. a** OAC cohort treatment options. **b** Plot summarizing features of 115 patients including overall survival (months), patient status, DOCTOR trial details, day of surgery, day of recurrence, overall stage, allocated treatment, available data of WGS, RNA sequencing and methylation profiling with distinct colour codes shown in the key. Patients are ordered by overall survival (months). **c** Kaplan–Meier plot for overall survival (log-rank test) comparing patients with different clinical stage (Stage I *n* = 12, Stage II *n* = 62, Stage III *n* = 35 and Stage IV *n* = 6). PET, positron emission tomography; CF, Cisplatin and 5-Fluorouracil; DCF, CF and docetaxel; RT, radiotherapy; WGS, whole genome sequencing. Source data are provided as a Source data file.

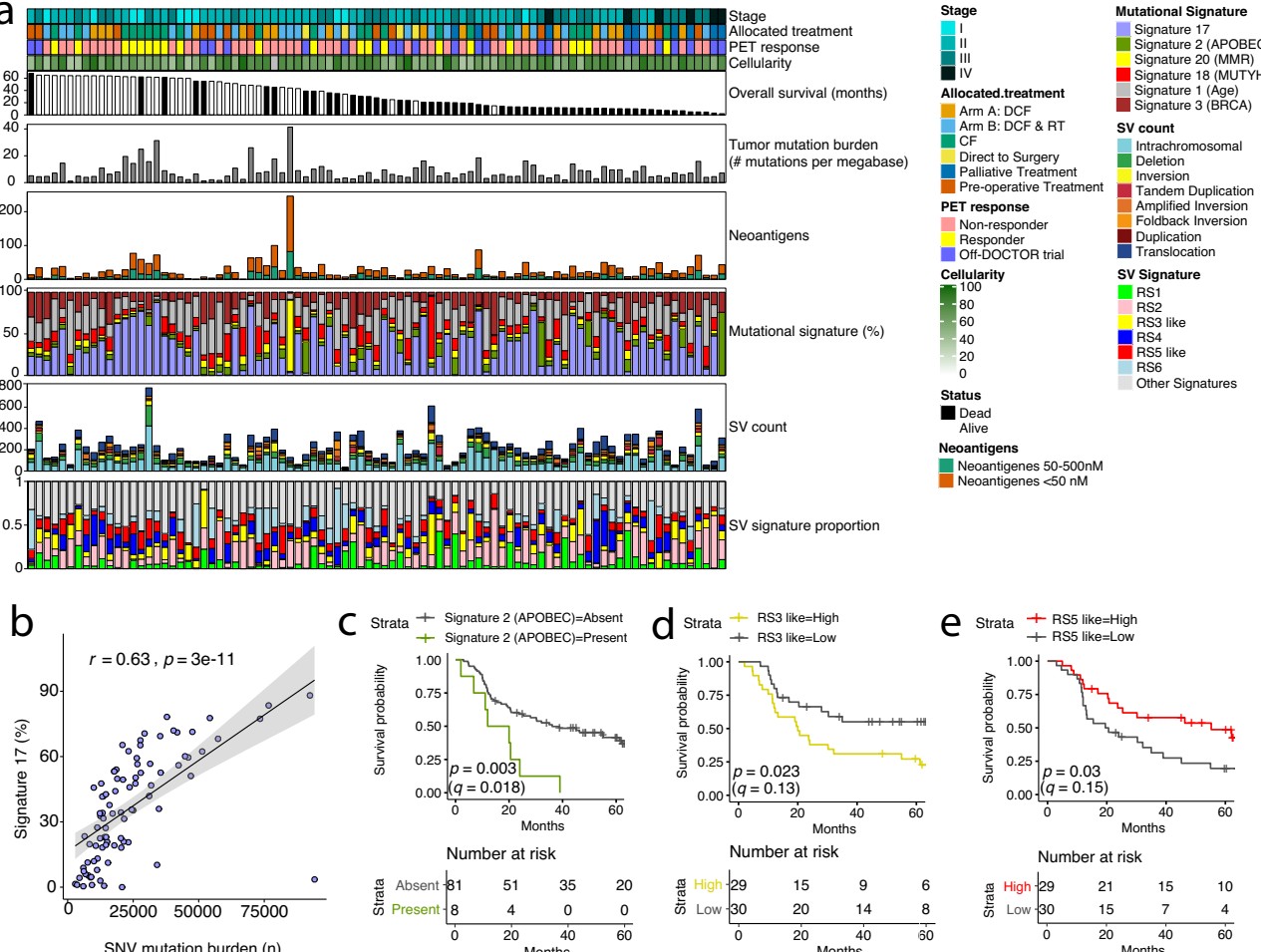

**Fig. 2 | The genomic landscape and mutational signatures in OAC. a** Bar plots displaying the genomic features of OAC samples within the cohort, with samples sorted by overall survival. The colour bar above the figure represents from top to bottom: stage, allocated treatment, PET response and tumour cellularity. The histograms from top to bottom are: overall patient survival with white bars are patients who are alive and black who are dead, the mutations per megabase, the number of neoantigens, the proportion of mutational signatures from SNV, the number and type of SVs, and the proportion of structural variant signatures. Clinical features per sample are annotated above the bar plots. **b** Pearson correlation (two-sided) of percent of mutational signature 17 burden (y-axis) with the total number of SNVs (x-axis) in each tumour ($n = 89$ biologically independent samples). Shading indicates 95% confidence intervals. **c**–**e** Kaplan–Meier plots of overall patient survival (log-rank test) with the number of patients (biologically independent samples) in each group shown in the table below each plot. Samples are stratified by the prevalence of **c** the APOBEC mutational signature (present ≥15% in a sample $n = 8$ and absent <15% $n = 81$), **d** RS3-like structural variant signature with samples stratified into low (lower tertile, $n = 30$) and high (upper tertile, $n = 29$) groups and **e** RS5-like structural variant signature with samples stratified into low (lower tertile, $n = 30$) and high (upper tertile, $n = 29$) groups. PET, positron emission tomography; CF, Cisplatin and 5-Fluorouracil; DCF, CF and docetaxel; RT, 45 Gy radiotherapy. Source data are provided as a Source data file.

correlation $r = 0.73$, $p = 2.84e{-}16$) (Supplementary Fig. 1a). Significantly mutated gene (SMG) analysis identified *TP53*, *CDKN2A* and *ARID1A*, which are known driver genes in oesophageal cancer[9,19], and other previously reported[8–11] driver genes (Supplementary Fig. 1b). Taken together, these data support that our cohort is representative of oesophageal adenocarcinoma.

The presence of SMGs was not associated with histopathological response or overall survival. Furthermore, we did not validate a previous report[20], of histopathology response being associated with somatic copy number changes in *CSMD1*, *ETV4*, *SMURF1* and mutations in *SMARC4* and amplification of *KRAS* and *GATA4* being associated with overall survival.

### Mutational signatures in oesophageal adenocarcinoma

SNV mutational signature analysis identified six signatures all previously described in OAC[11], signature 17 was the prominent signature in 48 out of 89 samples (Fig. 2a) and was associated with a higher SNV mutation burden (Fig. 2b). Signature 17 has been previously associated with 5-FU

treatment in a variety of cancer types, however, the samples herein were obtained before treatment. One sample (OESO_0118), was dominated by signature 20 (associated with mismatch repair deficiency), and harboured a pathogenic somatic mutation in *MSH2* (NM_000251.2:c.970 C > T; p.Gln324*). This sample contained the highest number of somatic mutations ($n = 94{,}195$) and neoantigen load, which suggest potential indications for immunotherapy. Signature 18, which is associated with *MUTYH* germline variants[21], was the dominant signature in a sample (OESO_0119) with a pathogenic somatic variant and loss of heterozygosity in *MUTYH* (NM_001128425.2:c.1213 C > T; p.Pro405Leu).

The mean number of somatic structural variants (SVs) was 231 (range 35–781) (Fig. 2a), consistent with previous reports[9]. SV signature analysis identified nine potential signatures at varying amounts within each sample (Supplementary Fig. 2a, b). Using cosine similarity, six of the signatures were most similar to RS1, RS2, RS3, RS4, RS5 and RS6 previously described[22] (Fig. 2a and Supplementary Fig. 2c). The signatures RS3 and RS5 have been associated with homologous recombination deficiency (HRD)[23,24]. We termed the two signatures RS3-like and

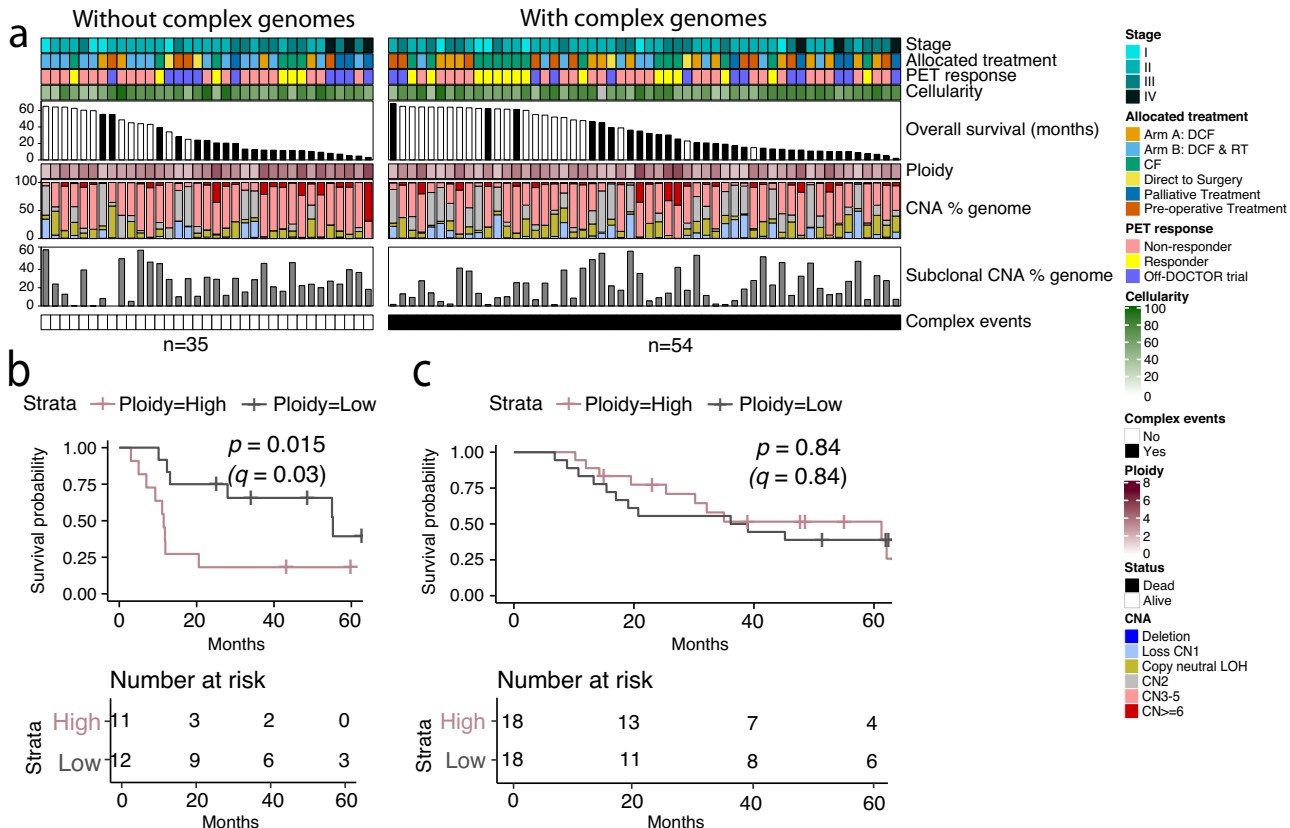

**Fig. 3 | Copy number alterations in OAC genomes. a** The upper colour bars represent from top to bottom: overall stage, allocated treatment, PET response and tumour cellularity. The histogram is the overall patient survival, white indicates patients who are alive and black who are dead. The ploidy is shown for each sample. The percentage of copy number alterations across each genome and the percentage of sub-clonal CNA are shown. Samples are grouped into those that contained complex genome events (black, *n* = 54) and those that did not (white, *n* = 35), and sorted by overall survival in descending order. There are 89 biologically independent samples. **b** Kaplan–Meier plot (log-rank test) of patient overall survival for tumours without complex events (*n* = 35) stratified as ploidy high (upper tertile, *n* = 11) or low (lower tertile, *n* = 12). **c** Kaplan–Meier plot (log-rank test) of patient overall survival for tumours with complex events (*n* = 54) stratified as ploidy high (upper tertile, *n* = 18) or low (lower tertile, *n* = 18). PET, positron emission tomography; CF, Cisplatin and 5-Fluorouracil; DCF, CF and docetaxel; RT, 45 Gy radiotherapy; CNA, copy number alteration. Source data are provided as a Source data file.

RS5-like as cosine similarities (0.65 and 0.79) suggested they are not classic signatures identified previously in breast cancer. Using HRDetect[23] and HRD scores[24] we identified nine samples that were predicted HRD by both approaches (Supplementary Fig. 2d). Interestingly, the HRD samples did not harbour germline variants classified as likely-pathogenic or pathogenic (in ClinVar) in HR-related genes[25]. One of the HRD samples (OESO_0047) did contain a somatic *BRCA1* missense change, however, an HR-proficient tumour (OESO_0118) also contained a frameshift *BRCA1* mutation (predicted pathogenic) and 3 other HR-proficient samples contained somatic *BRAC1* or *BRCA2* missense changes (see Supplementary Data 3 for all somatic coding mutations in the cohort). To further investigate potential perturbation of HRD genes, we also investigated *BRCA1* methylation levels in the cohort and did not identify any tumour with hyper-methylation of the promoter region of the *BRCA1* gene. We also compared *BRCA1* methylation levels in eight HRD and 61 non-HRD samples with available methylation profiles and did not identify any association between HRD and methylation changes in the *BRCA1* gene region. Taken together, these findings suggest the high TMB and large number of SVs may be linked to perturbed DNA repair that is not classical BRCA-driven HRD.

### Mutational signatures are associated with patient survival

We identified three mutational signatures associated with OS. Patients with an absent APOBEC signature (<15% of the signature load in a tumour) had better OS than the patients with an APOBEC signature present (log-rank, *p* = 0.003; adjusted for six mutational signatures q = 0.018) (Fig. 2c). The APOBEC signature was significantly higher in stage III patients compared to stage I (Supplementary Fig. 3a), however, the association with OS remained significant after adjusting for stage (Cox regression, *p* = 0.008) (Supplementary Fig. 3b). The APOBEC signature was also present in TCGA samples (Supplementary Fig. 4a). The pattern of OS in TCGA was similar to our cohort, with absent APOBEC signature tended to have better OS, although this was not significant (Supplementary Fig. 4b). The differences between our cohort and TCGA may be due to the differences in OS between the WGS and TCGA cohorts (Supplementary Fig. 4c).

The rearrangement signatures RS3-like and RS5-like were also associated with OS, where a high RS3-like signature was associated with worse OS (log-rank, *p* = 0.023) (Fig. 2d), and a high RS5-like signature was associated with better OS (log-rank, *p* = 0.03) (Fig. 2e). The presence of RS3-like and RS5-like signatures was not associated with tumour stage (Supplementary Fig. 3a). These results may suggest that the patients with a high RS5-like signature benefited from their treatment. However, adjusting the log-rank *p*-values for seven SV signatures suggested the possibility of false positives or heterogeneity for the association of RS3-like (log-rank, *q* = 0.13) and RS5-like (log-rank, *q* = 0.15) signatures with OS and the need for further analyses.

### Copy number aberrations and complex genomic events in oesophageal adenocarcinoma

Oesophageal cancers are driven by large-scale copy number aberrations (CNA) and complex genomic rearrangements. We identified

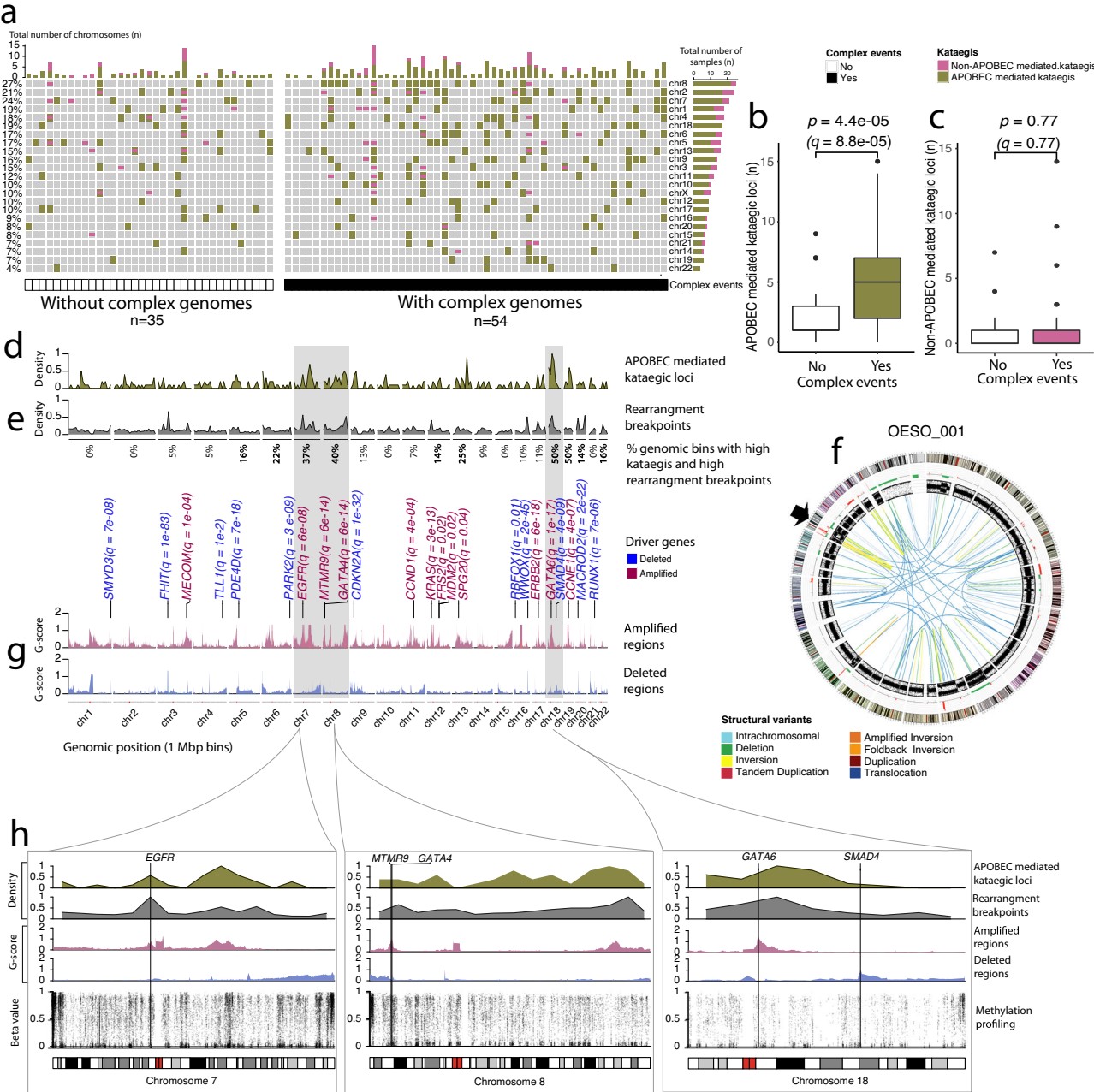

**Fig. 4 | Complex genomic events in OAC. a** Genomic location of non-APOBEC (pink) and APOBEC-mediated kataegic (olive green) loci per patient ($n = 89$ biologically independent samples) with chromosomes sorted by the number of kataegic loci. Samples are grouped into whether they did (black, $n = 54$) or did not contain complex genome events (white, $n = 35$), and sorted by overall survival in descending order. **b** Box plot of number of APOBEC mediated kataegic loci in tumours with ($n = 54$) and without ($n = 35$) complex events. **c** Box plot of number of non-APOBEC mediated kataegic loci in tumours with ($n = 54$) and without ($n = 35$) complex events. Box plots in **b** and **c** show the median values with the interquartile range (lower and upper hinge) and ±1.5-fold the interquartile range from the first and third quartile (lower and upper whiskers), $p$-values from Wilcoxon rank-sum two-sided test. **d–g** Genome-wide data for patients with APOBEC-mediated kataegic loci ($n = 77$ biologically independent samples) showing **d** density plot of kataegic loci, **e** density plot of rearrangement breakpoints with values below the plot indicating the percent of 1 Mbp genome bins overlapping kataegis and SV

breakpoints. Chromosomes highlighted grey harbour the most significant recurrent regions with co-localised kataegic loci and rearrangement breakpoints. **f** Circos plot of tumour OESO_001 highlighting rearrangement breakpoints and kataegic loci on chromosome 18 (arrow). Outer to inner panels: Chromosome banding, copy number alterations (green (below the line) represents loss and red (above the line) represents gain), BAF and somatic structural variants. **g** Recurrent focal amplifications (red) and deletions (blue) identified using GISTIC. OAC driver genes within focal events are shown ($q < 0.05$). Amplified or deleted regions with a G-score >0.12 from GISTIC are plotted. **h** Data shown for chromosomes 7, 8 and 18 in $n = 77$ biologically independent samples with APOBEC-mediated kataegic loci, from top to bottom: Density plot of kataegic loci, density plot of rearrangement breakpoints, recurrent focal amplification (red) and deletion (blue) events, rainfall plots of median methylation beta values (for $n = 69$ samples with methylation). Source data are provided as a Source data file.

tumours with complex events including chromothripsis, BFB or other clustered events, and found that 54 of 89 tumours contained at least one complex structural event (Fig. 3a and Supplementary Data 2). The mean ploidy within our samples was 3.01 (range 1.59−5.26), with most samples harbouring a high percentage of copy number gain within their genomes (Fig. 3a). The copy number events resulted in the deletion and amplification of known oesophageal cancer driver genes (Supplementary Fig. 5). In tumours that did not harbour complex events, high ploidy was associated with worse OS (log-rank, $p = 0.015$, $q = 0.03$) (Fig. 3b, c) and late tumour stage (Pearson $r = 0.41$, $p = 0.014$) (Supplementary Fig. 6a, b). Across all samples, intra-tumour heterogeneity was observed with an average of 24.5% sub-clonal CNAs (Fig. 3a), and no significant difference between tumours with or without complex events. The sub-clonal events were distributed across the genome (Supplementary Fig. 7a). Patients with a lower sub-clonal CNA percentage tended to have a better PET response (Wilcoxon rank-sum, $p = 0.032$) (Supplementary Fig. 7b).

Tumours harbouring complex events (Supplementary Fig. 8a) compared to tumours without complex events (Supplementary Fig. 8b) contained significantly more kataegic loci (Wilcoxon rank-sum, $p = 1.4e−04$) (Supplementary Fig. 8c). The number of kataegic loci within tumours was not associated with the overall tumour stage (Supplementary Fig. 8d) or OS between samples with or without complex events (Supplementary Fig. 8e−h). Further characterization revealed that the number of APOBEC-mediated kataegic loci was significantly higher in tumours with complex events (Wilcoxon rank-sum, $p = 4.4e−05$, $q = 8.8e−05$) (Fig. 4a, b), while no significant differences were found for non-APOBEC-mediated kataegic loci (Wilcoxon rank-sum, $p = 0.77$, $q = 0.77$ Fig. 4c). An average of 4.60 kataegic loci per sample (range 0−20) was identified, with over 20% of samples harbouring kataegic loci on chromosome 8, 2 and/or 7 (Fig. 4d). The kataegic loci are sometimes co-localised with SV breakpoints (Fig. 4e). To identify genomic regions with co-localised events in a non-random fashion, we estimated the frequency of genomic bins containing APOBEC-mediated kataegic loci and SV breakpoints more than the mean of normal distribution for both events. Here we considered a chromosome enriched with co-localised events if co-localised frequency was higher than the mean frequency of 13%. This identified chromosomes 7, 8, 18 and 19 with the highest enrichment of co-localized events (Fig. 4d, e), with specific tumours harbouring one or more of these events (Fig. 4f and Supplementary Fig. 9).

We further identified the most recurrent regions with co-localised events by estimating the combined percent inferred from kataegic loci frequency (Fig. 4d) and co-localised frequency (Fig. 4e) identified in our cohort. The combined percent suggested that chromosomes 7, 8 and 18 contained the most significant recurrent regions harbouring complex events and APOBEC-mediated kataegic loci (Supplementary Data 4). These regions harbour driver genes including *EGFR*, *GATA4*, *GATA6*, *MTMR9* and *SMAD4* that were significantly affected by copy number changes with GISTIC analysis (Fig. 4g). The methylation data was used to determine if aberrant methylation was associated with the recurrent regions containing complex events, kataegis and amplifications (Fig. 4h). We did not find differences between the methylation levels in those chromosomes or in the promoter regions and gene-body of the genes mapped to the recurrent regions when comparing tumours with and without complex events (Supplementary Figs. 10−14).

## Immune clusters in oesophageal adenocarcinoma are linked to survival

We investigated the proportions of immune cells in the tumour microenvironment (TME) of OAC samples ($n = 68$, $n = 45$ with matching WGS data) with RNA-seq data using two approaches, CIBERSORTx[26] and Consensus[TME] (Supplementary Data 5). Unsupervised clustering using GSVA scores from Consensus[TME]

transcriptomic deconvolution of 18 immune cell types revealed four clusters of samples (Fig. 5a and Supplementary Fig. 15a−d). Cluster 1 was enriched with immune cell infiltrate suggesting an immune hot TME. Cluster 2 was enriched with macrophages and myeloid-derived cells but depleted of lymphocytes, suggesting an immune-suppressed TME. Cluster 3 was enriched with moderate levels of lymphocytes and depleted of other immune cell infiltrates. Cluster 4 was immune cold as it lacked all immune cells including lymphocytes and myeloid-derived cells. The four RNA-seq clusters were also detected in 78 OAC samples from TCGA (Supplementary Fig. 15e−h).

The estimated cell type proportions from RNA-seq using CIBER-SORTx (Fig. 5b) were consistent with Consensus[TME]. We also investigated the proportions of 10 immune cell types in the TME using the methylation data ($n = 24$) with methylCIBERSORT[27] (Fig. 5c). There was a positive correlation between cell type estimates from methylation and transcriptomic data for B-cells (Pearson Correlation, $r = 0.67$, and $p = 3e−04$), neutrophils (Pearson Correlation, $r = 0.57$, $p = 0.0038$) and CD8 T-cells (Pearson Correlation, $r = 0.48$, and $p = 0.017$) (Supplementary Fig. 16a−c). However, monocytes, NK-cells and T-regulatory cells did now show significant correlations between methylation and transcriptomic data possibly due to distinct signature matrices (Supplementary Fig. 16d−f). CD4 T-cells and Eosinophils were zero in most estimations of methylation data.

The four clusters were associated with OS (adjusted log-rank, $q = 0.003$) and PFS (adjusted log-rank, $q = 0.00009$), with the immune suppressed cluster (Cluster 2) showing the worst survival and the immune hot cluster (Cluster 1) showing the best survival (Fig. 5d). In TCGA data there were a similar proportion of samples assigned to each cluster (Supplementary Fig. 17a), and although not significant, a similar trend in OS was observed in TCGA data although it was not significant for all clusters (Supplementary Fig. 17b).

Differential gene expression analysis comparing each cluster to the remaining clusters was performed to identify characteristic genes. Cluster 1 (immune hot), associated with better survival, contained decreased expression of 24 genes ($q < 0.05$) and increased expression of 244 genes ($q < 0.05$) with the top genes being the immune genes *CCL5*, *NKG7*, *GZMA*, *GZMB*, *GBP5*, *CD8A*, *LAG3* and *IDO1* (Fig. 5e). Cluster 2 (immune suppressed), associated with the worst survival, demonstrated 151 down-regulated and 382 up-regulated genes ($q < 0.05$) with the top genes associated with the extracellular matrix and EMT markers such as *SPP1*, *MMP3* and *COL4A1* (Fig. 5f). We performed gene set enrichment analysis (GSEA) using Hallmark gene signatures to identify signalling pathways in each cluster. Cluster 1 (immune hot, better survival) was uniquely enriched with interferon alpha/gamma responses, TNF-alpha signalling via NFKB, IL2/STAT5 pathway, apoptosis, hypoxia and p53 pathway, and showed lower levels of cell cycle and progression pathways such as E2F targets and G2M checkpoint (Fig. 5g). Cluster 2 (immune suppressed, worse survival) was uniquely enriched with cell cycle and progression pathways such as E2F targets, G2M checkpoint, and represented a lower level of p53 pathway (Fig. 5h). Both Cluster 1 and 2 showed activation of the KRAS signalling pathway and epithelial-mesenchymal transition (EMT). Cluster 3 (immune moderate) uniquely demonstrated lower levels DNA repair pathway. Cluster 3 also showed a higher level of p53 pathway and lower levels of immune responses as well as apoptosis (Supplementary Fig. 18a). Cluster 4 (immune cold) was enriched with metabolism pathways such as fatty acid metabolism and was high in protein secretion (Supplementary Fig. 18b).

To further investigate the clusters, immunohistochemistry (IHC) was performed using an antibody to CD8. We found a high correlation of CD8 protein expression with the *CD8A* gene (Pearson Correlation, $r = 0.82$, and $p = 1e−08$) (Fig. 5i) and confirmed Cluster 1 had the highest expression of CD8 and Cluster 3 had moderate expression (Fig. 5j). We also investigated the inter-tumour heterogeneity of CD8 T-cell populations in tumours from the four immune clusters by

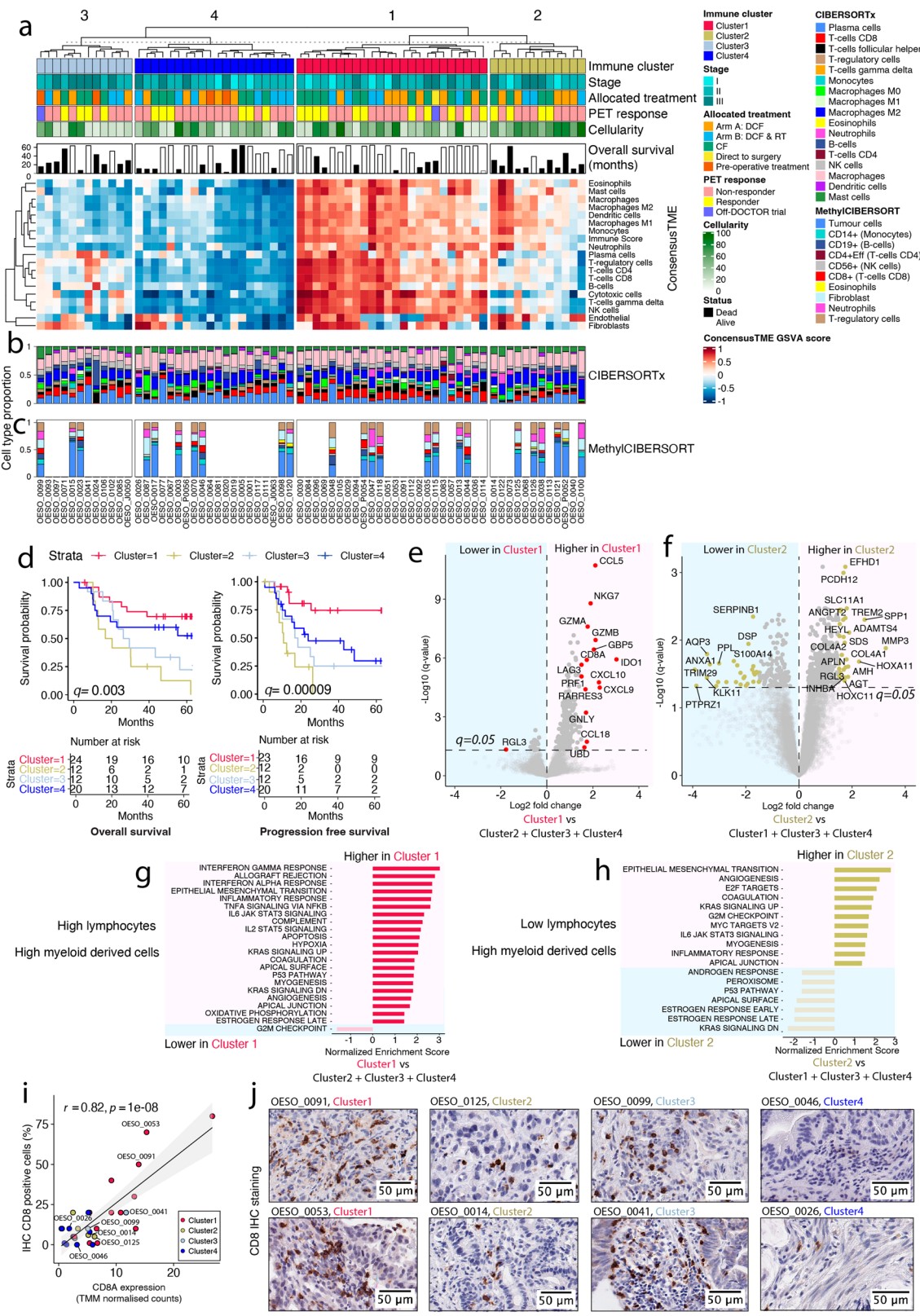

assessing whole slide IHC images of CD8. Tumours in Cluster 1 had an overall staining intensity of score 3+, however, the distribution of CD8 T-cells within the TME from these tumours may vary across the tumour tissue with low to strong staining in different parts of the tumours (Supplementary Fig. 19). Cluster 2 tumours have CD8 staining that varies from negative to low in different regions of each tumour, with few areas that are enriched with high CD8 T-cell

populations and an overall intensity score 1+ (Supplementary Fig. 20). Cluster 3 tumours display intra-tumour heterogeneity with regions showing negative to strong CD8 T-cell staining with the overall intensity score 2+ (Supplementary Fig. 21). Cluster 4 tumours are negative for CD8 T-cell staining with the overall intensity score 0 (Supplementary Fig. 22). These results suggest intra-tumour heterogeneity of CD8 T-cell population in Cluster 1 (immune hot),

**Fig. 5 | Immune microenvironment predicts patient outcome and future precision immunotherapy candidates. a** K-mean unsupervised clustering of patients ($n = 68$ biologically independent samples) using GSVA scores from ConsensusTME transcriptomic deconvolution of 18 immune cell types. The upper colour bars represent from top to bottom: immune cell cluster, overall stage, allocated treatment, PET response and tumour cellularity. The histogram is the overall patient survival, white indicates patients who are alive and black who are dead. **b** Estimated cell type proportion for the 68 patients using transcriptomics data and CIBERSORTx. **c** Estimated cell type proportion for patients ($n = 24$) with methylation profiling using MethylCIBERSORT. **d** Kaplan–Meier plot of overall survival (OS; left plot) and progression-free survival (PFS; right plot) using log-rank test to compare the four immune subtypes. The number of patients in each group is indicated below the plots. **e** Volcano plot of differential gene expression analysis comparing Cluster 1 with other immune clusters. Genes with absolute log scale fold change >1.5 and $q < 0.05$ are shown in red. **f** Volcano plot of differential gene expression analysis comparing Cluster 2 with other clusters. Genes with absolute log scale fold change >1.5 and $q < 0.05$ are shown in yellow. **g, h** GSEA normalised enrichment scores (x-axis) and enriched pathways (y-axis) using the genes differentially expressed in Cluster 1 (**g**) compared to other clusters and Cluster 2 (**h**) compared to other clusters. Represented pathways had an adjusted $p$-value <0.05 with GSEA, the pink background represents up-regulated pathways, the blue background down-regulated pathways. **i** Pearson correlation (two-sided) of the percentage of CD8 positive cells detected using immunohistochemistry (IHC; y-axis) and *CD8A* gene expression from RNA-seq (TMM; x-axis) for $n = 32$ biologically independent samples. Shading indicates 95% confidence intervals. **j** CD8 IHC was performed for $n = 32$ samples biologically independent samples, immunohistochemical staining for two representative samples from each of the four immune subtypes is shown (representative samples are highlighted in panel **i**). Images scanned at ×40 magnification, scale bar represents 50 μm. PET, positron emission tomography; CF, Cisplatin and 5-Flurouracil; DCF, CF and docetaxel; RT, 45 Gy radiotherapy; GSEA, gene set enrichment analysis. Source data are provided as a Source data file.

Cluster 2 (immune suppressed) and Cluster 3 (immune moderate), but the overall CD8 intensities were consistent with the estimated CD8 T-cell population from RNA-seq deconvolution analyses.

Clinico-pathological features of each immune cluster suggested higher nodal involvement in Cluster 2 (immune suppressed) compared to Cluster 1 (Kruskal–Wallis test, $p = 0.026$) (Supplementary Fig. 23a), while no stage biases were found for this subtype. Although not significant, lower pathological responses were found in Cluster 2 (immune suppressed) compared to all other clusters (Supplementary Fig. 23b). Immune cell composition in each cluster was further investigated using the ratio of neutrophils/T-cells (CD4+ and CD8+) across tumours. Accordingly, Cluster 2 (immune suppressed) and Cluster 4 (immune cold) showed a higher ratio of neutrophils/T-cells (CD4+ and CD8+) compared with Cluster 1 (immune hot) and Cluster 3 (immune moderate) tumours (Kruskal–Wallis, $p = 0.00013$) (Supplementary Fig. 23c). This supports a previous finding of an association of high CD8:CD163 ratio in OAC, comparable with Cluster 1 in our study, with an improved disease-free survival[28].

We next examined whether genomic features were associated with the four immune clusters in 45 samples with available RNA-seq and WGS. Strikingly, we found that the samples with rearrangement signature RS6, linked to clustered SVs, were enriched in Cluster 2 (two-sided binomial test, $p = 0.0185$; expected frequency 16%) (Supplementary Fig. 24a). Although not significant, we also found a higher percentage of kataegic loci in Cluster 2 compared to all other clusters (Supplementary Fig. 24b). Finally, Cluster 3 (immune moderate) showed lower Signature 17 compared to other clusters. (Supplementary Fig. 24c). We found the tumour mutation burden was lower than the mean 8.35 mutations/Mb (range 3.1–6.2 mutations/Mb) in Cluster 3 tumours.

## Oesophageal adenocarcinoma and the tumour microenvironment

We investigated bulk tumour expression ($n = 68$) of immune checkpoint molecules including *CTLA4, PD1, TIM3, LAG3, PDL1, PDL2, CD27* and *TIGIT* in the context of the four immune clusters (Kruskal–Wallis, $p = 0.00032$, $p = 9e{-}07$, $p = 4.6e{-}07$, $p = 8.8e{-}08$, $p = 0.00031$, $p = 4.4e{-}05$, $p = 5.9e{-}05$, $p = 2.6e{-}05$) (Supplementary Fig. 25a–h). Antibodies for these immune checkpoint molecules have been tested for other cancer types in the clinical trials. Cluster 1 has better survival and a high presence of lymphocytes and myeloid-derived cells demonstrated by high levels of immune checkpoint molecules which suggests a potential response to ICB immunotherapy. Cluster 2 has an immune-suppressed phenotype and is associated with complex rearrangements, which may contribute to the immune-suppressed phenotype. Cluster 2 showed moderate to high expression levels of immune checkpoint molecules with the highest expression for *TIM3* which suggests they could potentially be eliminated with ICB immunotherapy. Cluster 3 showed moderate OS, contained moderate lymphocytes in their TME and had a moderate expression of immune checkpoint molecules. This suggests potential benefits from ICB immunotherapy but with lower degrees compared to Cluster 1 and Cluster 2. Cluster 3 also contained a lower percentage of mutation signature 17. Finally, Cluster 4 was marked by low immune infiltrate and low expression of immune checkpoint molecules suggesting potentially no benefit from ICB immunotherapy. Together, our results reveal the presence of four candidate clusters of OAC tumours characterized by the TME and linked to patient survival (Fig. 6).

## Discussion

With advances in treatments, the understanding of the molecular features of OAC tumours and how these influence patient survival and treatment selection is essential. Previous studies established that OAC tumours with complex genomes are more aggressive and frequently diagnosed at later stages[9,29,30]. We explored whether WGS of OAC may indicate who will respond to neoadjuvant therapy and additionally if we could identify possible treatment strategies for those with poorer responses. We characterized 115 pre-treatment OAC tumours mostly from the DOCTOR phase II clinical trial using genomics, transcriptomics and methylation profiling, and stratified patients according to prognostic outcomes. The DOCTOR trial showed an early metabolic response (EMR) on PET scan to Cisplatin/5-FU chemotherapy in OAC was associated with a favourable survival outcome[17]. In non-EMR patients, the addition of docetaxel and radiation therapy improved PFS and OS in line with EMR patients. By combining genomic, transcriptomic and methylation profiles from the DOCTOR trial treatment arms, we show that OAC tumours contain genomic markers associated with good patient survival outcomes and propose four immune subgroups that could potentially predict response to immunotherapy in the future.

Oesophageal tumours may contain a large amount of intratumour heterogeneity with frequent genome gains[9,11]. We found that PET responders contained a lower percentage of sub-clonal copy number alterations, which suggests that tumours with less genomic heterogeneity responded better to treatment. Moreover, patients with tumours that did not contain complex genomes but were high tumour ploidy had worse survival. Recently, it was proposed that polyploidy in oesophageal adenocarcinoma originates from mitotic slippage caused by defective chromosome attachments[31], our findings suggest this may be a mutational mechanism that is a feature of late-stage oesophageal cancer, and warrants further investigation. Other genomic mutational features associated with patient survival were the AID/APOBEC mutational signatures and the RS3 and RS5 rearrangement signatures, and may point to potential mechanisms of OAC tumorigenesis. Consistent with previous reports[32]. A higher proportion of APOBEC signatures was detected in stage III compared to stage I tumours, however, the

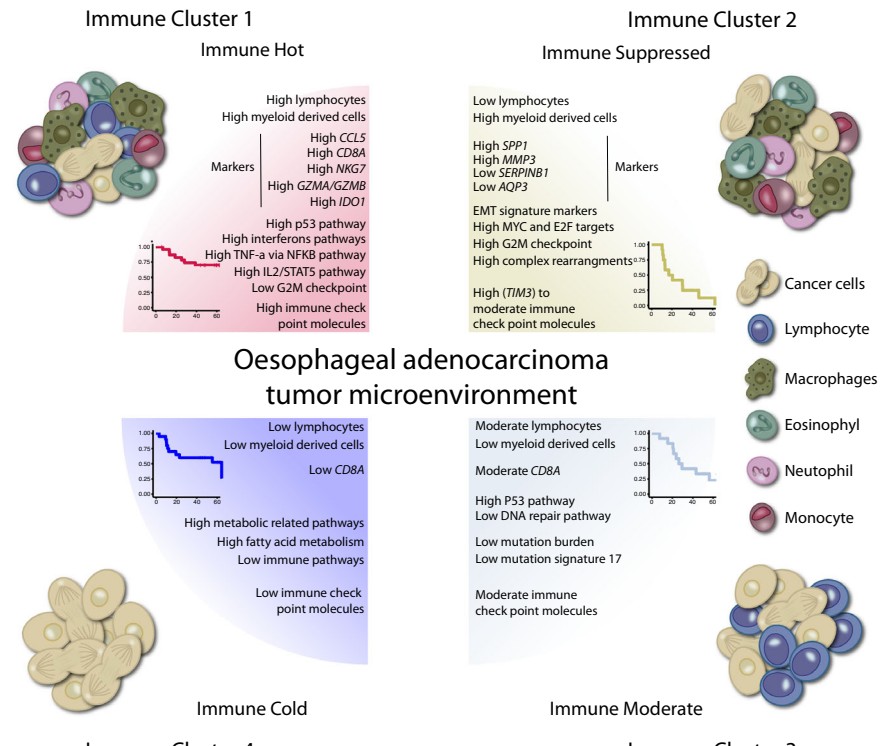

**Fig. 6 | Proposed Clusters present in oesophageal adenocarcinoma based on the tumour microenvironment.** Transcriptome analysis of oesophageal adeno-carcinomas identified four clusters of tumours based on the prediction of immune cells present within the tumour microenvironment. These clusters were associated with patient survival and tumour genomic features.

APOBEC signature correlated with worse OS even after adjusting for stage. The AID/APOBEC family enzymes induce endogenous muta-genesis through cytidine deaminase DNA-editing activity, although the factors responsible for activating mutagenesis remain unclear[33]. The mechanism of APOBEC-mediated mutagenesis are thought to be 'epi-sodic mutagenesis', where mutations can be generated in short bursts of activity followed by long periods of inactivity[34]. Somatic AID/APO-BEC-associated mutations have been linked to clusters of local hypermutation, termed kataegis, and have been observed in multiple cancer types[35,36] including OAC as well as pre-cancerous Barrett's oesophagus[14]. Co-localization of APOBEC-mediated kataegis and genomic breakpoints have been linked to complex events, in support of this we identified an enrichment of APOBEC-mediated kataegic loci occurring in tumours with complex events. Our analyses, indicated the APOBEC-mediated kataegic loci associated with SVs is contributing to tumorigenesis by impacting recurrently altered driver genes in OAC, in particular those on chromosome 7, 8 and 18 such as *EGFR*, *MTMR9*, *GATA4*, *GATA6* and *SMAD4*. However, the timing or relationship between the complex structural events and APOBEC-mediated muta-genesis is still unclear. Methylation profiling did not suggest an asso-ciation between methylation changes and complex events at *EGFR*, *MTMR9*, *GATA4*, *GATA6* and *SMAD4* genes regions. However, aberrant methylation has been previously linked to tumours with more SV events and tumours with focal amplifications[37].

Future studies focusing on ATAC-seq or direct DNA long read sequencing to resolve complex events and simultaneously profile methylation may provide insights into the mechanisms surrounding the complex genomic events and methylation patterns. Little is known about the contribution of complex genomic events to treatment response. However, APOBEC-mediated mutagenesis has been linked to innate immunity, and it has been proposed that tumours harbouring APOBEC mutation signatures may benefit from immunotherapy[34,38].

Furthermore, APOBEC enzymes may themselves present potential therapeutic targets in OAC[39].

SV signature analysis suggested six known rearrangement sig-natures in OAC genomes[22] including RS3-like and RS5-like signatures. RS3 and RS5 signatures have been linked to *BRCA1* aberrations and HRD[22]. However, pathogenic somatic or germline mutations in HR-related genes were lacking and we found few samples considered as HRD according to HRDetect/HRDscores, indicating that PARP inhibi-tors may be ineffective and that an alternate DNA repair may be per-turbed. Interestingly, the RS5-like signature was higher in patients with better overall survival, however, we are unable to conclude if this is due to tumours with an RS5-like signature responding to DNA damaging agents (cisplatin and 5-Fluorouracil) or whether other factors such as the possibility of RS5-like tumours being less aggressive may be linked with the better response. Therefore, further investigation would need to be conducted to confirm the patient survival effect of the RS5-like signature.

Future studies utilising patient-derived organoids (PDO) or patient-derived xenografts (PDX), could verify the genomic features associated with treatment response and clinical outcomes, including the number the sub-clonal copy number alterations and the presence of the AID/APOBEC or RS5-like mutational signatures. Previous studies have generated PDO from OAC tissues that histologically recapitulate the originating tissue sample[40–42]. Genome sequencing has shown a general concordance of mutations in OAC driver genes between the PDO and originating tissue samples, with few mutation differences[40,42], and multiple sub-clones have been identified in PDO models high-lighting tumour intra-heterogeneity within OAC samples[42]. PDO can be used to measure treatment response as they may recapitulate the clinical/pathological response seen in the patient to treatment[40,41], including responses to cisplatin and FOLFOX. In recognition of the value of PDO models a trial, registered in ClinicalTrials.gov

(NCT03283527), is enrolling oesophageal cancer patients with the aim to create a PDO biobank to assess the response to chemotherapy. Excitingly, several approaches for PDO generation have also been developed to preserve features of the TME[43,44], including tumour-infiltrating lymphocytes, which represent a platform to study the impact of the different immune clusters on immunotherapy response.

The tumour microenvironment impacts tumour development and response to treatment, therefore understanding the TME and how it shapes or is shaped by genomic features within tumour cells is critical. We identified four distinct clusters in OAC linked to the immune cells within the TME and associated with OS and PFS. Immune Cluster 1 (Immune Hot) was enriched with lymphocytes (i.e. CD4+ and CD8+ T cells) as well as myeloid lineage cells (i.e. macrophages, monocytes, and dendritic cells) and had the best OS and PFS of all the clusters. This is consistent with previous studies that associated immune hot tumours with prolonged survival in other cancer types[45–47]. It has been suggested that possessing the memory phenotype of lymphocytes can prolong the survival of patients[48]. The immune hot tumours in TCGA OAC cohort did not have significantly better survival compared to immune Cluster 4 (Immune Cold) tumours, possibly due to differences in survival between TCGA and the WGS OAC cohort, and/or the effectiveness of 5-FU in our cohort compared to TCGA OAC cohort who are not treated with neoadjuvant therapy.

The 5-FU was a shared therapeutic agent in the DOCTOR trial treatments and 5-FU has been reported to eliminate myeloid-derived suppressor cells in immune hot context[49]. Cluster 1 (immune hot) tumours were marked by high expression of immune genes such as *CCL5, NKG7, GZMA, GZMB, GBP5, LAG3, CD8A, CXCL9, CXCL10* and *IDO1*. Previous studies reported *CCL5, NKG7, CXCL9* and *CXCL10* as biomarkers for better survival in oesophageal squamous carcinoma and other cancer types[50,51], and high expression of *CD8A, GNLY, CCL5, CXCL9* and *CXCL10* in pre-treated OAC tumours with high T-cell infiltrates was linked to complete response to neoadjuvant chemoradiotherapy[28].

Similar to previous reports in gastroesophageal adenocarcinomas[52] the immune hot tumours in Cluster 1 were enriched with cytokine and immune-related pathways such as interferon alpha/gamma, EMT as well as p53 pathway. While Cluster 1 tumours showed lower levels of cell cycle and progression pathways such as the G2M checkpoint. The higher expression of immune checkpoint molecules, including *LAG3*, in the immune hot tumours may suggest potential benefits of ICB immunotherapy.

Cluster 2 (Immune Suppressed) tumours were depleted of lymphocytes but enriched with myeloid-derived cells. Immune-suppressed tumours showed the worst survival among all tumours in both our and TCGA OAC cohorts. Intriguingly, immune-suppressed tumours were not associated with stage but showed invasive and EMT phenotypes such as angiogenesis, G2M checkpoint, MYC and E2F targets activations. These tumours had a high number of dysregulated genes, a high expression of immune suppression markers (i.e. SPP1) and the highest number of nodal involvements after surgery. SPP1 (Osteopontin) is secreted by tumour cells and myeloid cells and is associated with aggressiveness and immune suppression in the TME[53] and has been reported as enriched in pre-treated OAC tumours which did not respond to neoadjuvant chemoradiotherapy[28].

Interestingly, we found a higher number of clustered rearrangements in immune-suppressed tumours, which agrees with previous reports that tumours with complex genomes are more aggressive and have lower responsiveness to therapies[9,54]. Moreover, the immune checkpoint molecules demonstrated moderate to high expression and the highest was observed for *TIM3* which suggests this potentially as the best ICB immunotherapy target for immune-suppressed tumours. The identification of the immune-suppressed cluster demonstrates the importance of developing treatments, potentially against EMT, MYC,

E2F targets or with immunotherapy, exploiting complex genomic events, to overcome the immune-suppressed environment.

Immune Cluster 3 (Immune Moderate) was depleted of myeloid lineage cells and macrophages but showed moderate levels of lymphocytes. Cluster 3 tumours were marked by higher p53 pathway level, lower DNA repair pathways, lower mutation burden and a lower proportion of mutation signature 17. The aetiology of signature 17 remains unclear, but in our cohort, it may be caused by oxidative damage[19] during digestion or chronic reflux as opposed to treatments with 5-FU[16], as the OAC tumours in our cohort were collected prior to any treatment. Therefore, the lower levels of signature 17 within Cluster 3 could potentially be explained by the lower levels of error-prone DNA repair pathways, or by an environment with less oxidative damage. In support of previous reports[55], we found a correlation between neoantigens and SNV mutation burden. Although we did not identify a higher neoantigen burden in Cluster 1 (Immune Hot), we did see a lower SNV mutation burden in Cluster 3 which was linked with moderate infiltrations of lymphocytes. Cluster 3 represented moderate expression immune checkpoint molecules and suggests a moderate benefit of ICB immunotherapy in the future.

Immune Cluster 4 (Immune Cold) was depleted of any immune cell types or immune and inflammatory related pathways, but they were enriched with metabolic pathways including fatty acid metabolism. The low expression of immune checkpoint molecules and dysregulated metabolic pathways in Cluster 4 suggest patients with immune cold tumours would not benefit from ICB immunotherapy but metabolic enzyme targets and small molecules could possibly be an effective treatment strategy in future. Using a multi-omic analysis of patients treated with current therapies, we confirm that a favourable immune landscape is required for long-term survival[56], and present data that provides a basis for understanding OAC responses to ICB. Data from Checkmate-577[57], Keynote-590[4], Keynote-180[5] and Keynote-181[58] demonstrating benefit for ICB in the adjuvant and metastatic setting make these data timely. However, in general, the majority of OAC patients do not respond adequately to immunotherapy, and to date, robust biomarkers of response have been elusive[4,5,58]. Programmed death ligand 1 (PDL1) is a biomarker to select patients with gastroesophageal cancer for ICB immunotherapy, however, this marker has not been validated for OAC[59]. The OAC immune clusters reported here may indicate potential responses to immunotherapy or neoadjuvant therapy. Combination therapy approaches, such as immunotherapy, chemotherapy and/or radiotherapy, are emerging and may demonstrate greater therapeutic benefits for patients. A challenge remains to better characterise immune evasion seen in the majority of OAC patients[57]. Future work to study samples pre and post-treatment to investigate the immune clusters, and spatial impact of cells within the TME using single-cell and spatial transcriptomics will provide more insights into the interplay between the tumour and the microenvironment during treatment.

In summary, we investigated genomic, transcriptomic and methylation features of OAC tumours from patients who were part of the phase II clinical trial using neoadjuvant therapy. We proposed genomic prognostic features including mutational signatures, rearrangement signatures, copy number alterations, clonality, and complex genomic events. We identifying distinct immune clusters that correlate with OS and PFS of pre-treatment neoadjuvant therapy and simultaneously predicting the potentials of ICB immunotherapy. We introduced genomic and transcriptomic biomarkers for the immune clusters that potentially facilitate stratifying patients for selected therapy in the future. Markedly, we characterized genomic and transcriptomic features of the poor prognosis patients Cluster 2 (Immune Suppressed) which helps to predict responses to neoadjuvant therapy and also develop better selected therapy, monotherapy, or combination therapy with chemotherapy and/or radiotherapy in the future.

## Methods

### Sample cohort, DNA and RNA extraction

This study includes OAC samples ($n = 91$) from patients recruited through DOCTOR clinical trial (ANZCTR - ACTRN12609000665235). Approval has been granted by the Metro South Health Research Ethics Committee (HREC/2020/QMS/62117), the University of Queensland Research Ethics Committee (UQ2020/HE001913), and the QIMR Berghofer Research Ethics Committee (P3559). Additional OAC samples ($n = 24$) were obtained through the Cancer Evolution Biobank (HREC/10/QPAH/152, UQ/2011001287). All participants ($n = 115$) provided written informed consent. Sex and/or gender of participants was not considered in the study design; most participants were male (80 of 89) which is expected for OAC. Sex of participants was determined based on self-report. We did not perform a sex analysis, as our aim was to look for markers relevant in all OAC samples.

OAC patients from the AGITG DOCTOR clinical trial received pre-operative chemotherapy and/or radiotherapy. They were treated with cisplatin and 5-fluorouracil (CF), and all were assessed by PET scan on Day 14. Patients who showed EMR went through second CF. Non-responders were randomized for two arms including CF and docetaxel (DCF, Arm A) or DCF concurrent with 45 Gy radiotherapy (DCF and RT, Arm B). Patients who were off-DOCTOR trial received preoperative treatments, palliative treatments, or went directly for surgery.

For each patient, a tumour and a matched normal sample were available. The normal sample was obtained from the buffy coat or normal tissue adjacent to the tumour site. Tumours were frozen in RNAlater for downstream genomic analysis. DNA and RNA were extracted using the Qiagen AllPrep DNA/RNA mini kit according to the manufacturer's protocol (Qiagen, Germany). Buffy coat DNA was extracted using the Qiagen Flexigene kit according to the manufacturer's protocol (Qiagen, Germany). An additional tumour specimen was also formalin-fixed paraffin-embedded (FFPE). Hematoxylin and eosin (H&E) slides were assessed for tumour content by an anatomical pathologist. Clinical follow-up information was precisely recorded for maximum 5.5 years and is shown in Supplementary Data 1.

### Whole-genome sequencing and SNP array

Prior to WGS, the tumour content of DNA samples was assessed using SNP array analysis using the Omni 2.5–8, V1.0−Illumina BeadChips according to manufacturer's instructions (Illumina, San Diego, CA, USA). Tumour cellularity (tumour content) of each sample was estimated with the qpure tool[60] (v1.0.0). Samples with a tumour cellularity >~40% were selected for WGS which was performed using the HiSeqX-Ten (Illumina, San Diego, CA, USA) at either Macrogen (Geumcheon-gu, Seoul, South Korea) or The Kinghorn Cancer Centre, Garvan Institute of Medical Research (Sydney, Australia). The sequence data for 22 samples in the off-DOCTOR trial cohort were reported previously[9]. After sequencing, adaptors were trimmed with Cutadapt (v1.9) and reads were mapped to human reference genome GRCh37 using BWA-MEM (v0.7.15) and SAMtools[61] (v1.9) Duplicated read were marked with Picard MarkDuplicates (https://broadinstitute.github.io/picard) (v2.8.15). The mean coverage was estimated for samples using qCoverage tool (v0.7) (https://github.com/AdamaJava/adamajava). Tumour and matched normal samples had a median coverage of 67.38 (range 48.00−80.66) and 33.77 (range 23.97−39.20), respectively. All samples demonstrated tumour purity >26% using ascatNGS[62] (v4.0.1) (median purity 53%).

### Somatic SNV and indel calling

Somatic SNVs were called using a dual calling approach via consensus of two distinct tools, qSNP[63] (v2.1.4) and GATK HaplotypeCaller[64] (v4.0.4.0). Indel calling was performed using GATK[64] (v4.0.4.0) and variant annotation using gene consequences was implemented using SnpEff[65] (v2.1.2).

### Mutational signature analysis

Mutational signatures in WGS and TCGA samples were identified using a non-negative matrix factorization (NMF) method[33]. Mutational signatures retrieved from NMF analysis were compared to known mutational signatures v2 in COSMIC database using cosine similarity (http://cancer.sanger.ac.uk/cosmic/signatures, applied April 2020). The contribution of each signature across samples was estimated using a quadratic programming approach available in the R package SignatureEstimation. To avoid over-fitting, signatures <10% for each sample were omitted and mutations were reassigned to the remaining signatures.

### Neoantigen prediction

Class I HLA genotypes were computed for tumour-normal pairs of WGS using Optitype (v1.3.1) with default parameters. Predicting neoantigens was performed using pVAC-Seq[66] (v4.0.10) pipeline with default parameters and binding affinity was estimated using NetMHCpan (v4.0). Variants were annotated for wild-type and mutant peptide sequences as recommended through variant effect predictor (v86) (VEP) from ENSEMBL. We considered epitopes with binding affinity Inhibitory Concentration (IC50) ≤500 nM as potential neoantigens that bind to HLA alleles. Epitopes with strong binding affinity were considered those with IC50 of <50 nm. We prioritized and identified expressed neoantigens with an IC50 ≤500 nM using qbasepileup tool (https://github.com/AdamaJava/adamajava/tree/master/qbasepileup). Individual samples were run in SNP mode through qbasepileup in order to count reference genome base and mutant bases at each SNP position in aligned RNA-seq BAM. Duplicate and poorly mapped reads were excluded and a mutation was considered to be expressed if there was a minimum of 10 reads with evidence of the mutation.

### Significantly mutated gene analysis

We performed a consensus approach using multiple tools to identify significantly mutated genes (SMGs) affected by SNVs/indels. These tools were OncodriveFML[67], MutPanning (v2.0) and MutSigCV, and were run with default parameters. We execute OncodriveFML using CADD v1.0 via the web interface (http://bbglab.irbbarcelona.org/oncodrivefml/home) and MutPanning through the available module in GenePattern (https://www.genepattern.org/modules/docs/MutPanning). We used the threshold $q$-value <0.1 for OncodriveFML[67] and MutSigCV[68], and FDR <0.05 for MutPanning[69]. We reported the gene as significant if it was considered significant in two or more tools.

### Structural rearrangement variants, copy number alterations

Structural variants (SV) were identified using qSV[70] (v0.3). SV breakpoints and potential consequences of rearrangements are called using ENSEMBL annotation (v75) of known genes. All COSMIC Cancer census genes were downloaded (https://cancer.sanger.ac.uk/cosmic), and tier1 and tier2 genes were selected. Additionally, any known driver genes[8–11] were included in our analysis. Copy number calling was executed on sequencing data using ascatNGS[62] (v4.0.1). To annotate copy number status, we categorized copy number alterations (CNA) to copy number loss (copy number 1), copy number gain or amplification (copy number ≥6) and homozygous deletion (copy number 0). CNAs at the gene level were identified through annotating according to ENSEMBL known genes (v75). Recurrent CNAs were identified for chromosomal regions using GISTIC2.0[71]. A gene considered significant if affected by focal copy number changes with a confidence level >0.95 and a $q$-value <0.05. Only significant driver genes identified through SMG analysis and/or reported previously[8–11] were chosen for further investigation.

### Structural rearrangement signature and clustering

We used a similar approach to mutational signature analysis and applied NMF to identify structural rearrangement signatures[33]. SVs

were assigned categories as previously described[22]. Essentially, SVs were initially classified into: deletions, inversions, duplications and inter-chromosomal translocations, then SVs were further annotated according to size and whether the breakpoints were clustered or non-clustered. We considered different-sized groups for all events except translocations, and these groups were: 1–10 kb, 10–100 kb, 100 kb–1 Mb, 1–10 Mb, and more than 10 Mb. Clustered SV breakpoints were identified using the BEDtools cluster function[72]. Cluster events were defined as the occurrence of ≥10 breakpoints in a 1 Mb genome bin as reported by Letouze et al.[73].

### Homologous recombination deficiency analysis

We estimated levels of homologous recombination deficiency (HRD) using two distinct approaches scarHRD R package[74] (https://github.com/sztup/scarHRD) and HRDetect[23] (https://github.com/eyzhao/hrdetect-pipeline). The scarHRD package uses next-generation sequencing data including WGS and WES to calculate levels of telomere allelic imbalance (tAI), loss of heterozygosity (LOH) and the number of large-scale transitions (LST). scarHRD computes the unweighted numeric sum of LOH, tAI, and LST reported as HRD-sum[74], and represents HRD score in this study. The HRDetect package is a WGS and WES-based classifier predicting *BRCA1* and *BRCA2* deficiencies. The package uses SNV signatures, rearrangement signatures and HRD index including tAI, LOH and LST scores. The package computes a HRDetect score, a probability value quantifying the degree of BRCA1/BRCA2 defectiveness. In this study, we classified samples as HRD with an HRD score >42 and HRDetect score >0.7.

### Sub-clonal copy number alterations

We identified sub-clonal copy number alterations from whole genome sequencing using Battenberg R package[35]. All parameters were set on default mode. GRCh37 reference files required for running Battenberg were downloaded as recommended (https://github.com/Wedge-Oxford/battenberg). All required files were generated as recommended (https://github.com/cancerit/cgpBattenberg). An ignore file was generated to include chromosomes 1-22. In the output, segmental solutions, *p*-values and fractions were listed from A (the most confident solution) to F (the least confident solution). We selected segments with significant copy number alterations FDR <0.05. The proportion of sub-clonal copy number alterations for each chromosome and whole genome were calculated for each sample through the following approach: (a) Sub-clonal CNA of individual chromosome: We summed the length of significant segments with copy number alterations located at each chromosome and divided by length of the chromosome. (b) Sub-clonal CNA of whole genome: We summed the length of all significant segments with copy number alterations across the genome and divided by the length of the genome. Proportions were multiplied by 100 to calculate the percentage.

### Complex events and kataegis

Localized complex events and chromothripsis were identified using ShatterSeek R package[54] (in https://github.com/parklab/ShatterSeek). ShatterSeek identifies chromothripsis using copy number alterations and structural variants. We prepared structural variants from qSV (v0.3) and copy number inputs from ascatNGS (v4.0.1) as instructed. We identified high-confidence calls as previously described[54], and to avoid false positive calls in quiet genomes we set higher thresholds compared to Isidro Cortés-Ciriano et al.[54]. These thresholds include clusters of interleaved SVs >10, oscillations between two or three copy-number states >6, *q*-value <0.05 for chromosomal enrichment and an exponential distribution of breakpoint test, and *q*-value <0.2 for the fragment joins test. We additionally included localized complex events identified with clusters of interleaved SVs >30 and an exponential distribution of breakpoint test *q*-value <0.05. Finally, we performed an in-depth manual curation to remove false positives. Samples with more than one complex event were considered as complex genomes (Supplementary Data 2).

Localized hypermutations (kataegis) were detected using SeqKat R package[75,76] (https://cran.r-project.org/web/packages/SeqKat/index.html). Seqkat uses a sliding window approach to test the deviation of SNV trinucleotide content and also an inter-mutational distance from the expected chance. SeqKat performs an exact binomial test to evaluate whether the proportion of each of the 32 tri-nucleotides within each window is higher than expected. The resulting *p*-values were then adjusted using FDR. We used default parameters in this study. SeqKat computes a hypermutation score and AID/APOBEC-mediated kataegis score separately for each window as described here (https://cran.r-project.org/web/packages/SeqKat/index.html). We downloaded trinucleotide count file and chromosome length file for GRCh37 reference genome from https://cran.r-project.org/web/packages/SeqKat. We estimated the percentage of genomic bins with APOBEC kataegic loci and rearrangement breakpoints more than the mean of normal distribution for both events across the genome (refer to Fig. 4). We also examined in which chromosomes this percentage is higher than the mean percentage across the genome (highlighted the percentages with bold font in Fig. 4). To identify chromosomes the most enriched by APOBEC kataegic loci and rearrangement break-points, we estimated the frequency of APOBEC kataegic loci overlapping with rearrangement breakpoints through multiplying: (a) frequency of kataegic loci for each chromosome, (b) percentage of genomic bins with high APOBEC kataegic loci and rearrangement breakpoints; and then ranked the combined percent from high to low (refer to Supplementary Data 4).

### Methylation profiling

Whole-genome DNA methylation profiles from 72 samples are included in this study (Fig. 1 and Supplementary Data 1). Methylation data for 31 samples assayed by methylation Illumina 450 K arrays was downloaded and processed (accession number: GSE72874)[13]. We performed methylation profiling for an additional 41 samples with sufficient DNA (performed at Macrogen) using 0.7 µg of DNA and the Illumina EPIC arrays.

### Methylation analysis of complex genomic events and Homologous recombination deficiency

To investigate methylation patterns surrounding complex genomic events we analysed DNA methylation profiles of 69 samples with matched WGS (Supplementary Data 1), 31 of the samples were assayed with methylation Illumina 450 K arrays (accession number: GSE72874)[13] and 38 samples with the EPIC array. The raw idat files for the 450 K and EPIC array data were imported using minfi[77] and probes were removed if the detected *p*-value was >0.01 or there were fewer than three beads in at least 5 % of samples using ChAMP[78] (v.2.12.4). A total of 442,767 common probes were identified between the two platforms (450 K and EPIC arrays). These probes underwent a β-mixture quantile normalization (BMIQ)[79] to account for probe type 1 and type 2 biases, followed by a quantile normalisation (QN). Further filtering was performed to remove: (1) probes in non-CpG sites; (2) X or Y chromosome; (3) single-nucleotide polymorphism-related polymorphisms as per Zhou et al.[80]; and (4) probes that map to multiple locations as per Nordlund et al.[81]. After filtering, 377,176 probes remained and were used in further analysis. A batch correction process was applied using removeBatchEffect from the limma package (v3.38.3). We performed gene-level methylation analysis for the genes identified within recurrent complex events that co-localised with kataegic loci or recurrent copy number changes, including *EGFR*, *MTMR9*, *GATA6*, *GATA4* and *SMAD4* located at chromosome 7, 8 or 18. In this analysis a ploidy correction was used to determine whether genes were amplified, whereby genes were considered amplified when ASCATploidy was <2.7 and the copy number for the gene was ≥6 or

ASCAT ploidy was ≥2.7 and the copy number for the gene was ≥9. We also investigated *BRCA1* methylation levels in samples with and without HRD. Probes mapped to these genes were evaluated for methylation levels according to their genomic features (complex events, recurrent copy number and kataegic events). We performed pairwise comparisons between samples with and without the events at the chromosome level for chromosomes 7, 8 and 18 (complex events in each sample are included in Supplementary Data 2). To investigate methylation and HRD, eight HRD samples and 61 non-HRD samples with available methylation profiles were included in the analysis.

### RNA sequencing and normalisation

RNA sequencing was performed on 79 RNA tumour samples with a RIN score (>5). Libraries were prepared from RNA using the TruSeq Stranded mRNA kit and sequenced with 100 bp paired-end reads. Sequencing data were aligned to human reference genome GRCh37 using STAR aligner[82] (v2.5.2a). Sequencing adaptors were trimmed using Cutadapt[83] (v1.9) and gene annotation, transcript and exon features of ENSEMBL (release 89) were used to compute counts of individual genes for all samples. Quality controls were assessed using RNA-SeQC[84] (v1.1.8) and gene expression was estimated using RSEM[85] (v1.2.30). According to the counts, gene annotations and sequencing depth of the samples, transcripts-per-million (TPM) was computed across samples. TMM (trimmed mean of M values) normalization was performed using R package edgeR[86]. Linear scales of TPM and TMM were used in analyses unless explained.

### RNA deconvolution of the tumour microenvironment, clustering and differential expression analysis

The cell types within the Tumour microenvironment (TME) were estimated for 68 samples from RNA sequencing data using Consensus[TME] R package[87] (https://github.com/cansysbio/ConsensusTME) and CIBERSORTx[26]. We initially assessed the robustness of cell type estimation for 79 samples (linear scale TPM normalized data) using CIBESRORTx, but of these only 68 samples showed statistical significance $p < 0.05$ for robust cell type estimation and were therefore included for TME analysis using Consensus[TME] (Supplementary Data 1). The purity of excluded samples ($n = 11$) was too high or too low and possibly the reason for statistical failure. Consensus[TME] performs a consensus approach for 18 different cell types that compiles gene sets from seven published TME cell estimation methods: CIBERSORTx, Bindea et al.[48], Danaher et al.[88], Davoli et al.[89], MCP-counter[90], TIMER[91] and xCell[92]. Consensus[TME] combines gene-set-based methods and regression-based methods using multiple statistical tests to infer cell-type enrichments. Consensus[TME] was run using linear scale TPM normalized data with the Gene Set Variation Analysis (GSVA) enrichment method[87]. GSVA is a non-parametric, unsupervised method for estimating the variation of gene set enrichment through expression data set and is ideal for standard analytical methods such as survival analysis and clustering[93]. We performed K-means clustering analysis on inferred GSVA scores for 18 cell types across the cohort. The optimal number of clusters was calculated through a consensus approach integrating different methods including elbow, silhouette, gap statistics, and Euclidean distances (general agreement of three out of four methods). The clustering was permuted ($n = 1000$) to find the most stable and robust clusters.

Differential gene expression analysis was performed to compare immune clusters. Effective library sizes were estimated by TMM method through edgeR[86] prior to voom function from the limma-voom R package (https://bioconductor.org/packages/release/bioc/html/limma.html). We performed differential analyses for each cluster versus other clusters and used the design matrix "-0+factor". The *p*-values were adjusted using Benjamini–Hochberg procedure.

Pathway enrichment analysis was performed using pre-ranked gene set enrichment analysis (GSEA) and Hallmark gene signatures from the Molecular Signatures Database (MSigDB, http://www.gsea-msigdb.org/gsea/msigdb/collections.jsp). The GSEA input was prepared by ranking the genes using log-fold changes from differential expression analysis. Pathways with adjusted *p*-value <0.05 were included in analyses.

### Methylation deconvolution of the tumour microenvironment

We used DNA methylation data for cell type deconvolution analysis. This analysis was restricted to 24 samples with available matched methylation profiling and RNA sequencing data from the same tissue biopsy and RNA data that showed statistical significance $p < 0.05$ for robust cell type estimation using Consensus[TME] (Supplementary Data 1). Six of the samples were assayed with methylation Illumina 450 K arrays (accession number: GSE72874)[13] and 18 samples were assayed using Epic array (accession number: GSE200707). We followed the normalisation process as described for analyses of methylation data for complex events. Here we identified 443,544 probes common between the two platforms (450 K and EPIC arrays), and after filtering 377,909 probes remained for further analysis. DNA methylation deconvolution was performed with the R package methylCIBERSORT[27] using the oesophageal cancer signature matrix provided in the methylCIBERSORT package. Cell type abundance was estimated using CIBERSORT[26].

### Validation of findings using TCGA data

To validate our findings, we analysed RNA sequencing data from the ESCA TCGA cohort. We downloaded the ESCA TCGA BAM files from https://portal.gdc.cancer.gov/projects/TCGA-ESCA using an approval from QIMR Berghofer Research Ethics Committee (P2095). We selected 78 samples of oesophageal adenocarcinoma subtype (classified as Esophagus Adenocarcinoma (NOS)) in the clinical information (cBio-Portal, https://www.cbioportal.org/). We performed RNA sequence analysis (sequence alignment, normalization and TME profiling) using the same processes as described above.

### Immunohistochemistry

IHC staining was performed to assess CD8 expression on 32 samples that had undergone RNA sequencing and for which FFPE tumour blocks were available. Staining was performed using an automated Roche Ventana Discovery Ultra® (Roche Diagnostics AG, Rotkreuz, Switzerland). The slides were placed in the stainer with primary anti-Human CD8 (Dako, #M7103, Clone: C8/144B, mouse monoclonal, diluted 1:4000) and secondary antibodies (Anti-mouse HQ and HQ-HRP) followed by DAB to develop the chromogenic staining. IHC slides were scanned at ×40 magnification on an Aperio ScanScope AT turbo Brightfield Slide Scanner (Leica Biosystems, Germany). Whole-slide images were captured as TIF images using Aperio ImageScope software (v12.3.3). IHC slides were assessed by a pathologist (SS) using a semi-quantitative approach to estimate the maximum number of positively stained lymphoid cells (cytoplasm/plasma membrane) per high power field. The overall intensity of whole-slide images of immune clusters was scored as negative (score = 0), low (score = 1+), moderate (score = 2+) and strong (score = 3+). Overall intensity was scored considering all sections of slides for each immune cluster. IHC TIF images were loaded into ImageJ, where a scale bar was included on each image.

### Statistical analysis

Statistical analyses were performed using R (v3.6.2). The significant threshold for *p*-values and false discovery rate (FDR) was less than 0.05 unless specified. Comparisons of three or more groups with continuous variables were performed using Kruskal–Wallis test. Pairwise comparisons were tested using Wilcoxon rank-sum test and multiple testing problems were removed by calculating FDR. All the Box plots show the median values with the interquartile range (lower and upper

hinge) and ±1.5-fold the interquartile range from the first and third quartile (lower and upper whiskers). Survival analyses were performed using a log-rank test in categorical comparisons and cox regression for continuous variables. Log-rank survival test for immune clusters was adjusted for treatment and stage using inverse probability weighting[94]. Multi-variate Cox regression model was applied to predict the correlation of APOBEC mutation signature and Overall Survival (OS) after adjustment for stage. A two-tailed binomial test was performed to test the overrepresentation of samples with present RS6 signature (>15%) in Cluster 2. All figures are prepared in R (v3.6.2).

### Reporting summary

Further information on research design is available in the Nature Portfolio Reporting Summary linked to this article.

## Data availability

The whole genome and RNA sequence data presented in this study is available at the EGA under accession number EGAS00001002864. The sequence data are generated from patient samples and therefore are available under restricted access. Data access can be granted via the EGA with completion of an institute data transfer agreement, and data will be available for a defined time period once access has been granted. Methylation array data used in this study are available from GEO [https://www.ncbi.nlm.nih.gov/geo/]. Methylation array data for 31 samples was downloaded from GEO under accession number GSE72874. Methylation data for an additional 41 samples is available from GEO under accession number GSE200707. Data generated in this study are provided in Supplementary data files. Exome and RNAseq data from TCGA for ESCA adenocarcinomas were downloaded from TCGA [https://portal.gdc.cancer.gov]. The sequence data is controlled access data which requires dbGaP Access. Source data are provided with this paper.

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

## Acknowledgements

This work was funded by the Australian National Medical and Research Council (NHMRC) (APP1147118). N.W. is funded by a National Health and Medical Research Council of Australia (NHMRC) Senior Research Fellowship (APP1139071) and Investigator grant (2018244). L.G.A. is supported by an NHMRC Early Career Fellowship (APP1109048). The Cancer Evolution Biobank is supported by the PA Research Foundation (RSS_2020_040). The AGITG DOCTOR Study was funded by a National Health & Medical Research Council (NHMRC) Project Grant (APP1011782). Sanofi Aventis Australia Pty Ltd (no grant number) provided docetaxel for the AGITG DOCTOR Study. We would like to acknowledge the support of the Estate of the late Alec Pearman and Ms Di Jameson. The results shown here are in part based upon data generated by the TCGA Research Network: https://www.cancer.gov/tcga. We are grateful to the patients that have participated in this project.

## Author contributions

M.M.N: Formal analysis, methodology, investigation, visualization, validation, writing—original draft. F.N., L.G.A., V.F.B., V.L. and S.B.: Formal analysis, investigation, writing—review & editing. K.P.: Resources, data curation, writing—review & editing. G.L.: Formal analysis, validation. L.T.K., V.A., O.H., C.L., S.W. and Q.X.: Software, formal analysis, data curation, writing—review & editing. O.K., R.L.J. and V.G.: Formal analysis, writing—review & editing. S.S.: Formal analysis, validation, writing—review & editing. J.T., E.W., G.T.M., S.P.A., J.M., M.B., R.F., C.K., J.S., L.N., R.B., K.W., E.B. and J.R.Z.: Resources. B.M.S., J.S. and T.P.: Resources, writing—review & editing. K.N.: Formal analysis, investigation, funding acquisition, writing—review & editing. D.I.W.: Resources, funding acquisition, writing—review & editing. J.V.P.: Software, resources, investigation, supervision, infrastructure, funding acquisition, writing—review & editing. A.P.B.: Conceptualization, resources, investigation, supervision, project administration, funding acquisition, writing—review & editing. N.W.: Conceptualization, resources, formal analysis, investigation, supervision, project administration, funding acquisition, writing—original draft.

## Competing interests

J.V.P. and N.W. are founders of genomiQa Pty Ltd, and members of its Board. All other authors have no competing interests.

## Additional information

[1]QIMR Berghofer Medical Research Institute, Brisbane, QLD 4006, Australia. [2]Frazer Institute, The University of Queensland, Woolloongabba, QLD 4102, Australia. [3]Princess Alexandra Hospital, Woolloongabba, QLD 4102, Australia. [4]Faculty of Medicine, University of Queensland, Brisbane, QLD 4006, Australia. [5]Anatomical Pathology, Australian Clinical Labs, 2153 Sydney, Australia. [6]Mater Research Institute, Mater Misericordiae, South Brisbane, QLD 4101, Australia. [7]Department of Medical Oncology, Calvary Mater Newcastle, Waratah, NSW 2298, Australia. [8]Department of Radiation Oncology, Calvary Mater Newcastle, Waratah, NSW 2298, Australia. [9]Royal Brisbane and Women's Hospital, Herston, QLD 4029, Australia. [10]Flinders University Department of Medical Oncology, Flinders Medical Centre, Adelaide, SA 5042, Australia. [11]Nepean Cancer Care Centre, Nepean Hospital, Sydney, NSW 2747, Australia. [12]Department of Medical Oncology, Royal Hobart Hospital, Hobart, TAS, Australia. [13]Department of General Surgery, Royal Hobart Hospital, Hobart, TAS, Australia. [14]NHMRC Clinical Trials Centre, University of Sydney, Sydney, NSW 2006, Australia. [15]Department of Epidemiology and Preventive Medicine, Monash University, Melbourne, VIC 3004, Australia. [16]Medical Oncology Unit, The Queen Elizabeth Hospital and University of Adelaide, Adelaide, SA 5011, Australia. [17]Flinders University Discipline of Surgery, Flinders Medical Centre, Adelaide, SA 5042, Australia. [18]These authors contributed equally: Andrew P. Barbour, Nicola Waddell. ✉e-mail: a.barbour@uq.edu.au; nic.waddell@qimrberghofer.edu.au

