## [Peer review file · Nature Communications]

REVIEWER COMMENTS

Reviewer #1 (Remarks to the Author): Expert in oesophageal cancer genomics, clinical research, and tumour microenvironment

The authors conduct a genomic/transcriptomic study of a recently completed clinical trial called DOCTOR. While it is meritorious to conduct molecular analysis of patient samples from clinical trials, there are a number of major concerns that mitigate enthusiasm.

1. The findings do not reveal new insights into the molecular pathogenesis of EAC, nor do they provide perspectives into mechanisms of resistance that could be useful for a new clinical trial.
2. Functional data are lacking. In other words, if there are immune clusters or sub clones that might confer resistance, then a hypothesis needs to be formulated and then tested experimentally. Model systems, such as 3D organoids, would have been useful.
3. The molecular analysis would be strengthened by consideration of epigenetic changes through use of ATAC-seq. Furthermore, single-cell ATAC-seq/RNA-seq would be useful to highlight tumor heterogeneity as a complex issue.

Reviewer #2 (Remarks to the Author): Expert in oesophageal cancer genomics and clinical research

Naeini et al performed combined genomic (WGS) and transcriptomic profiling on a cohort of treatment naïve esophageal adenocarcinomas in which post therapy (as part of the DOCTOR trial) patient outcome was known in order to identify potential determinants of overall survival. They identified several genomic features that were associated with overall survival; including APOBEC and RS3-like rearrangement signature, and increased copy number based sub-clonal heterogeneity associated with poor survival/worse response to therapy. Using the transcriptomic data, they were also able to classify the EACs into 4 immune categories and showed that the immune based clusters were associated with survival. Overall, this was a well performed study and the known details of treatments each patient received along with the combined WGS and transcriptomics is a major strength. One potential weakness of the study is that several similar recent manuscripts have been published that may take some away from the novelty/significance.

Comments:

1: For the cases with a high RS3-like signature, the authors state no alterations in BRCA1 or BRCA2 were identified. Were any alterations in other HR related genes identified (PALB2, RAD50,...)? If so, that could potentially explain the similarities while still being somewhat different.

2: For the survival analysis comparing APOBEC, while statistically significant, a HR of 1.01 to 1.1 is quite low and doesn't seem clinically relevant? Given how close it is to one, confirmation in an independent data set, such as the TCGA used later in the manuscript, would be valuable. The same is true for the RS-3 and RS-5 like signatures.

3: It looks like high ploidy also correlated with stage in the 'non complex event' group. Could ploidy and stage be related? (ie could more advanced cancers have time to develop higher ploidy or alternatively could high ploidy drive progression?)

4: The fact cluster 3 had a lower Sig 17 is interesting. Could this give any insight into the cause of this signature? (ie does cluster 3 have something uniquely missing compared to the other clusters that could induce Sig 17 like mutations?)

5: The authors should be a little careful between cause and effect vs associations in the discussion. The authors appropriately use qualifiers, but some statements may still need to be toned down. For example, line 287+, with the given data it is difficult to conclude that CF treatment improved survival because the RS-5 like signature is present. Just as likely, an alternative would be a biological process that is creating RS-5 causes less aggressive cancers or responds to CF treatment.

6: A major issue within the manuscript is the lack of comparison to a few other recently published studies (Jammula Gastro 2020, Goedegebuure Oncoimmunology 2021, de Klerk Mol Onco 2021, Derks Annals of Onco 2020) who performed (at least somewhat) similar experiments, although maybe less robustly. Nevertheless, it does somewhat call into question the novelty of the current study and as such some of the discussion should be devoted to compare the authors results to the prior work to stress the novel findings and show how the studies may complement each other.

6b: In Jammula et al, through bulk RNA-seq immune analysis they also identify an 'immune hot' molecular subtype. However, in this study the 'immune hot' subtype has the worst prognosis. Based on what Jammula et al reported, is there any obvious differences between the subtypes that might give clues on the importance of individual cell types on survival?

6c: Similar to the neutrophil/T-cell ratio used in this study, In Goedegebuure et al found that a high CD8:CD163 ratio was associated with improved disease-free survival. While the neutrophil/T-cell ratio was high in clusters 2 and 4, are the CD8/CD163 ratios similar or different in the clusters (ie does it better correlate with treatment response/survival)?

7: This reviewer appreciated the well described methods section. It served to increase the confidence in the results.

Minor:

1: Line 72-77, the authors may want to adjust the sentence to note that while signature 17 is associated with 5-FU treatment, in OAC something else is likely causing the signature. As the authors are aware, signature 17 is also the dominate signature in Barrett's oesophagus and OAC not exposed to 5-FU (as in this manuscript).

2: While being well defined in the methods, a quick ½ sentence on the definition of complex events in the results (line 155) would aid in reading if there is space.

3: For Figure S4, 'del' is defined as homozygous deletion correct?

4: Line 321, The manuscript shows enriched myeloid cells in cluster 2, but unsure if it shows MDSCs. It could have been missed, but if not, please be more specific on the cell types.

5: Line 522-532: Paragraph seems duplicated from the one above.

Responses to reviewers

We thank the reviewers for their helpful suggestions and comments. We have provided a responses to each of the comments from reviewers in our rebuttal document. When changes have been to the manuscript we have used track changes and provided both a tracked change and a clean version of the revised manuscript.

REVIEWER #1

1. The findings do not reveal new insights into the molecular pathogenesis of EAC, nor do they provide perspectives into mechanisms of resistance that could be useful for a new clinical trial.

RESPONSE: Our study aimed to identify features of pre-treatment tumours that are associated with survival (overall survival and progression-free survival) in patients who had received neoadjuvant therapy. Although our study did not specifically aim to provide insights into the molecular pathogenesis of EAC, we were still able to identify some novel findings We highlight the novelty and importance of our study as follows.

- Our study included treatment naïve tumours from oesophageal patients who received clinical trial-based neoadjuvant therapy. We used genomic, transcriptomic and methylation features assessed pre-treatment to stratify patients based on post-treatment outcomes (overall and progression-free survivals). To our knowledge, this makes the study unique in the clinical research context, and our findings will help to stratify patients in future clinical trials.
- Our whole genome analyses allowed us to identify genomic features associated with overall survival including the APOBEC mutational signature, the RS3-like structural variant signature and increased copy number-based sub-clonal heterogeneity.
- Our transcriptome analysis identified and verified immune cells within the tumour microenvironment and their associations with patient outcomes. Strikingly, we identify four immune clusters using unsupervised analyses that have unique cell type admixtures and are associated with overall and progression-free survival. This is a novel finding and suggests the microenvironment cell admixture could be used to stratify patients for future clinical trials of neoadjuvant therapy.
- The immune clusters in our study also inform the potential use of immunotherapy. Previous immunotherapy clinical trials in OAC¹⁻³ suggest some benefits for OAC, however, it has yet to revolutionize treatment for OAC. In our data, we showed that PDL1 is generally low expressed in OAC which may explain the failure of PDL1 treatment. In addition to indicating who may benefit from immunotherapy, the immune clusters we present may suggest alternative immunotherapy treatments including the immune checkpoint molecule *TIM3* which we found to be highly expressed in a subset of samples.
- Although not our primary focus we were able to reveal insights into OAC tumorigenesis. Complex genomic rearrangement events are known to play an important role in the pathogenesis of OAC. We show that APOBEC-mediated kataegic loci are associated with these events including those near known driver genes (including *EGFR*, *MTMR9*, *GATA6*, *GATA4* and *SMAD4*), this finding points to potential DNA repair mechanisms linked to breakpoints.

Furthermore, we hypothesized that the complex events are linked to aberrant methylation causing genome instability. To test this, we have included new methylation data to study the methylation status at chromosomes 7, 8 and 18 around the *EGFR*, *MTMR9*, *GATA6*, *GATA4* and *SMAD4* gene regions.

CHANGES TO MANUSCRIPT: We have altered the flow of the introduction and in particular the discussion. This rewrite of these specific sections has framed our work within the current literature and better highlighted the novelty of our findings.

2. Functional data are lacking. In other words, if there are immune clusters or sub clones that might confer resistance, then a hypothesis needs to be formulated and then tested experimentally. Model systems, such as 3D organoids, would have been useful.

RESPONSE: Our study aimed to identify potential genomic and transcriptomic features of treatment naïve tumours that are associated with post-neoadjuvant therapy outcomes (overall survival and progression-free survival). We did not aim to investigate mechanisms of resistance per se.

We agree that the next step would be to perform functional validation in suitable model systems. Unfortunately, we are unable to test the findings experimentally in our cohort. The tumour tissue samples within the cohort were collected into RNAlater and for the majority of samples, the tissue was exhausted during DNA and RNA extraction. Thus, our cohort is not suitable for the generation of preclinical models such as patient-derived xenografts or organoids.

Within the recent literature, there are a variety of organoid and patient-derived xenografts that have been described for OAC. The findings already reported for these models begins to validate some of our findings and future work with these models could be undertaken to confirm others.

CHANGES TO MANUSCRIPT: We revised the “Discussion” to include a paragraph highlighting the current and potential future use of these models: *“Future studies utilising patient derived organoids (PDO) or patient derived xenografts (PDX), could verify the genomic features associated with treatment response and clinical outcomes, including the number the sub-clonal copy number alterations and the presence of the AID/APOBEC or RS5-like mutational signatures. Previous studies have generated PDO from OAC tissues that histologically recapitulate the originating tissue sample⁴³⁻⁴⁵. Genome sequencing has shown a general concordance of mutations in OAC driver genes between the PDO and originating tissue samples, with few mutation differences^{43,45}, and multiple sub-clones have been identified in PDO models highlighting tumour intra-heterogeneity within OAC samples⁴⁵. PDO can be used to measure treatment response as they may recapitulate the clinical/pathological response seen in the patient to treatment^{43,44}, including responses to cisplatin and FOLFOX. In recognition of the value of PDO models a trial, registered in ClinicalTrials.gov (NCT03283527), is enrolling oesophageal cancer patients with the aim to create a PDO biobank to assess the response to chemotherapy. Excitingly, several approaches for PDO generation have also been developed to preserve features of the TME^{46,47}, including tumour-infiltrating lymphocytes, which represent a novel platform to study the impact of the different immune clusters on immunotherapy response.”*

3. The molecular analysis would be strengthened by consideration of epigenetic changes through use of ATAC-seq.

RESPONSE: Methylation changes have been widely reported in oesophageal cancer. Therefore, we have chosen to perform new analyses in the form of methylation analysis to consider epigenetic changes. The methylation work has been integrated into several parts of the manuscript including the analysis of complex genomic events and the analysis of the tumour microenvironment.

We looked at methylation patterns in samples identified with complex events co-localized with kataegic loci (**Figure 4e-g**) at chromosomes 7, 8 and 18. In most regions, methylation levels were similar in samples with and without complex events. We focused on driver genes with recurrent complex events including *EGFR*, *GATA4*, *MTMR9*, *GATA6* and *SMAD4* (**Figure 4i and Supplementary figure 10-14**), but did not identify methylation differences for these specific genes.

The methylation data was also used to investigate methylation levels in the *BRCA1* gene region to determine if tumours with homologous recombination deficiency contained methylation of the *BRCA1* gene (for details see response to comment 1 from reviewer 2).

We also used the methylation data for deconvolution to infer cell type admixtures within the tumour microenvironment. Methylation data from samples with matched transcriptomics data (n=24) was deconvolved using methylCIBERSORT. The methylation data supported the RNAseq work, as we found a positive correlation for estimated CD8 T-cells, B-cells, and Neutrophils between methylation and transcriptomics data. Notably, the identified immune clusters are mostly based on lymphocytes populations along with neutrophils and macrophages (**Figure 5, Supplementary figure 16**).

CHANGES TO MANUSCRIPT: To incorporate the methylation work we have substantially amended the text, figures and supplementary information. These changes fall into three categories:

1. Incorporation of the methylation data within the manuscript:

- We have amended the “Title” of the manuscript to refer to “Multi-omic” analysis rather than just genomic and transcriptomic.
- We amended the “Abstract” and “Introduction” to indicate methylation profiling was undertaken and included some background on methylation in OAC in the introduction on p.3: *“DNA methylation is a key component of epigenetic mechanisms for maintaining genome instability and regulating gene expression¹⁰ and distinct methylation subgroups with prognostic implications have been described in OAC¹¹.”*
- **Figure 1 and Supplementary Table 1** have been revised to include which samples have methylation profiling data.
- We have included new sections in the methods to describe the methylation analysis on p.29, the sections are: *“Methylation profiling”, “Methylation analysis of complex genomic events and Homologous recombination deficiency”, AND “Methylation deconvolution of the tumour microenvironment”.*

2. To determine whether methylation is linked to chromosome instability in OAC we performed a new analysis to determine if there were methylation changes at regions of complex chromosome changes or kataegis. This analysis did not show an association between methylation changes and complex events at

EGFR, GATA4, MTMR9, GATA6 and SMAD4 gene regions. We also investigated methylation levels in the *BRCA1* gene region and the potential associations with HRD. The changes to the manuscript are:

- We have split Figure 3 into two Figures (Figure 3 and Figure 4) to incorporate the methylation analysis within complex genomic regions. Specifically, we have added a new panel, **Figure 4i**, that shows methylation values and the density of genome breakpoints and APOBEC-mediated kataegic loci around recurrent regions with complex events on chromosomes 7, 8 and 18. We have also revised **Supplementary Table 2** to indicate samples with recurrent complex events.
- We have included new **Supplementary Figures 10-14** which are boxplots showing the methylation status at CpG sites within the gene body and around the transcription start site of key genes. The plots compare the methylation status of samples with and without complex events, kataegic events and recurrent copy number alterations. The comparisons include driver genes that harboured recurrent complex events on chromosomes 7, 8 and 18 (*EGFR, GATA4, MTMR9, GATA6* and *SMAD4*).
- We have added text in the Results section to include these new analyses: *“The methylation data was used to determine if aberrant methylation was associated with the recurrent regions containing complex events, kataegis and amplifications (Figure 4i). We did not find differences between the methylation levels in those chromosomes or in the promoter regions and gene-body of the genes mapped to the recurrent regions when comparing tumours with and without complex events (Supplementary Figure 10-14)”*.
- We added text in the Discussion section to discuss findings *“Methylation profiling did not suggest an association between methylation changes and complex events at EGFR, MTMR9, GATA4, GATA6, and SMAD4 genes regions. However, aberrant methylation has been previously linked to tumours with more SV events and tumours with focal amplifications⁴⁰. Future studies focusing on ATAC-seq or direct DNA long read sequencing to resolve complex events and simultaneously profile methylation may provide insights into the mechanisms surrounding the complex genomic events”*.

3. Deconvolution of the tumour microenvironment (TME) using methylation profiling for 24 samples with matching RNAseq. We have performed a new analysis using methylCIBERSORT (for methylation data) and CIBERSORTx (for RNA-seq data) to estimate the proportion of cells present in the TME. These analyses allowed us to compare the results from methylation and RNAseq. This new analysis has been incorporated into the paper as follows:

- We have updated the methods section “Tumour microenvironment, clustering and differential expression analysis” to include the CIBERSORTx RNAseq analysis.
- New **Figure 5b** which shows cell type proportions estimated using CIBERSORTx for 68 samples with RNAseq. This analysis was performed to complement the previous ConsensusTME analysis and enable a comparison to methylation deconvolution.
- New **Figure 5c** which shows cell type proportions estimated using methylCIBERSORT for 24 samples with matching RNAseq.
- New **Supplementary Figures 16** showing Pearson correlation plots of estimated cell types from transcriptomic (using CIBERSORTx) and methylation data using (methylCIBERSORT).
- We added text in the “Results” section to include these new analyses: *“The estimated cell type proportions from RNA-seq using CIBERSORTx (Figure 5b) were consistent with Consensus^{TME}. We*

also investigated the proportions of 10 immune cell types in the TME using the methylation data ($n=24$) with methylCIBERSORT⁴ (**Figure 5c**). There was a positive correlation between cell type estimates from methylation and transcriptomic data for B-cells (Pearson Correlation, $r=0.67$, and $p=3e-04$), neutrophils (Pearson Correlation, $r=0.57$, $p=0.0038$) and CD8 T-cells (Pearson Correlation, $r=0.48$, and $p=0.017$) (**Supplementary Figure 16a-c**). However, monocytes, NK-cells and T-regulatory cells did not show significant correlations between methylation and transcriptomic data possibly due to distinct signature matrices (**Supplementary Figure 16d-f**). CD4 T-cells and Eosinophils were zero in most estimations of methylation data.

- The four clusters were associated with OS (adjusted log-rank, $q=0.003$) and PFS (adjusted log-rank, $q=0.00009$), with the immune suppressed cluster (Cluster 2) showing the worst survival and the immune hot cluster (Cluster 1) showing the best survival (**Figure 5d**). In TCGA data there were a similar proportion of samples assigned to each cluster (**Supplementary Figure 17a**), and although not significant, a similar trend in OS was observed in TCGA data although it was not significant for all clusters (**Supplementary Figure 17b**)."

Furthermore, single-cell ATAC-seq/RNA-seq would be useful to highlight tumor heterogeneity as a complex issue.

RESPONSE: We have sought to address this comment on heterogeneity using two approaches: i) analysis of public single-cell RNAseq data and ii) Immunohistochemistry (IHC) staining of tumour samples used in our paper. Unfortunately, we are unable to include single-cell ATAC-seq data of our samples since this approach works best with fresh frozen samples, as the process requires a DNA cutting step⁵ that uses a hyperactive Tn5 transposase^{6,7} to simultaneously cut and ligate adapters for high-throughput sequencing. The clinical tissue samples included in this study are collected in RNAlater, and we only have remaining RNAlater tissue from a small number of patients, which together means ATAC-seq was not possible. However, as above we did include new data to profile the DNA methylation of the patients in our paper. A summary of the single cell data and IHC is as follows:

i) Analysis of public single-cell data

We downloaded public single cell RNAseq from SRX3245501 (<https://www.ncbi.nlm.nih.gov/sra/>)⁸ and re-analysed the data. We aligned the scRNAseq FASTQs using CellRanger (v3.0) (<https://support.10xgenomics.com/single-cell-gene-expression/software/pipelines/latest/what-is-cell-ranger>). The initial alignment and cell count quantification were performed on each sample library using CellRanger Counts (v2.1.0) to the CellRanger reference package hg19 (v1.2.0). Sample libraries were then aggregated using the CellRanger aggr (v2.1.0) tool with default normalise depth mode. Further analyses including quality control (QC) and normalization were performed using R (version 4.0.1) (<https://www.r-project.org/>) and Seurat R package (version 4)⁹ (<https://satijalab.org/seurat/>).

The original study from Wu et al. profiled two OAC samples (EAC01 and EAC02) and three ESCC samples and showed that the OAC samples contained a high tumour purity and as would be expected clustered separately from the ESCC samples. We focused our analysis on the two OAC samples to study the heterogeneity between and within these samples. Using UMAP and tSNE we identified two clusters of cells that correspond to the two OAC samples (**Rebuttal Figure 1a**) which suggest intra-patient heterogeneity. In agreement with Wu et al., we found that the two OAC samples have high tumour purities (70-80%) and therefore will have a low proportion of immune cells which does limit the use of

the data to study the tumour microenvironment. We used the data from the two OAC samples to investigate the expression of the markers we identified in the immune clusters within our paper (*CD8A*, *NKG7*, *GZMA*, *CCL5*, *SPP1*, *AQP3*, *GBP5*, *MMP3*, *COL4A1* and *CXCL9*, from **Figure 5**). Only *CCL5* in EAC01 and *AQP3* in EAC02 contained a high expression, with other markers having a low or no expression (**Rebuttal Figure 1b**). The expression of *AQP3* and *CCL5* was not uniform across all cells within patients, suggesting intra-patient heterogeneity. *AQP3* expression is from the tumour cells, therefore this demonstrates heterogeneous expression between tumours and within EAC02.

Rebuttal Figure 1: UMAP and tSNE analyses of two OAC samples that underwent scRNAseq (EAC01 and EAC02). We accessed and analyzed single-cell data that was first described in Wu et al. a) UMAP and tSNE visualization of the single data from two OAC patients: EAC01 (red) and EAC02 (blue). b) tSNE analysis biomarkers including CD8A, NKG7, GZMA, CCL5, SPP1, AQP3, GBP5, MMP3, COL4A1 and CXCL9 for identified immune clusters (figure 4) in this study.

However, due to the small number of samples and immune cells in this dataset, we are not able to robustly assess tumour heterogeneity. Therefore, we did not add these findings to the revised manuscript but have expanded on the discussion.

ii) Immunohistochemistry of samples in our paper

We performed whole slide immunohistochemistry (IHC) to investigate intra-tumour heterogeneity of CD8 T-cell populations in the four identified immune clusters. We show that CD8 T-cell populations are heterogeneous in Clusters 1, 2 and 3. However, all sections of Cluster 4 (Immune Cold) are mostly depleted from CD8 T-cells. We have included new data in the manuscript to show whole slide images and zoom-in images from distinct parts of each slide for identified immune clusters in **Supplementary Figures 19-22**.

The distribution of CD8 T-cells in each of the clusters is as follows:

- Cluster 1 tumours show “low-strong” staining of CD8 T-cells with overall intensity “clinical score 3+” considering distinct sections of the slides.
- Cluster 2 tumour sections are mainly enriched with “negative-low” CD8 T-cell populations but a few areas are enriched with a high CD8 T-cell population. This suggests the overall intensity “clinical score 1+”.
- Cluster 3 sections show “none-strong” CD8 T-cell populations with the overall intensity “clinical score 2+” for this cluster.
- Cluster 4 sections show “negative” CD8 T-cell populations with the overall intensity “clinical score 0” for this cluster.

These results suggest tumours have heterogeneous populations of CD8 T-cells, and that this staining shows intra-tumour tumour heterogeneity. With the overall intensities from IHC validating our findings from bulk transcriptomic data.

CHANGES TO MANUSCRIPT: To incorporate the IHC additional IHC work and discuss future approaches we have made the following changes:

- We added text to the Methods section to describe the whole slide IHC analysis: *“Whole-slide images were captured as TIF images using Aperio ImageScope software (v12.3.3). IHC slides were assessed by a pathologist (SS) using a semi-quantitative approach to estimate the percentage of positive CD8 cells and score IHC staining. Overall intensity of whole-slide images of immune clusters was scored as negative (clinical score=0), low (clinical score= 1+), moderate (clinical score=2+) and strong (3+). The overall intensity was scored considering all sections of slides for each immune cluster”*.
- We have included new **Supplementary Figure 19-22** showing representative whole-slide IHC images for the four immune clusters. These images highlight inter-tumour heterogeneity of CD8 staining.

- We have added text in the “Results” section explaining tumour heterogeneity: *“We also investigated the inter-tumour heterogeneity of CD8 T-cell populations in tumours from the four immune clusters by assessing the whole slide images of CD8 IHC. We found the distribution of CD8 T-cells within the TME from tumours in Cluster 1 can vary across the tumour tissue with low to strong staining in different parts of the tumours, with the overall intensity “clinical score 3+” (Supplementary Figure 19). Cluster 2 tumours have CD8 staining that varies from negative to low in different regions of each tumour, with few areas that are enriched with high CD8 T-cell populations and an overall intensity “clinical score 1+” (Supplementary Figure 20). Cluster 3 tumours display intra-tumour heterogeneity with regions showing negative to strong CD8 T-cell staining with the overall intensity “clinical score 2+” (Supplementary Figure 21). Cluster 4 sections show negative CD8 T-cell staining across tumour sections with the overall intensity “clinical score 0” (Supplementary Figure 22). These results suggest intra-tumour heterogeneity of the CD8 T-cell population in Cluster 1 (immune hot), Cluster2 (immune suppressed) and Cluster3 (immune moderate), but the overall CD8 intensities were consistent with the estimated CD8 T-cell population from RNA-seq deconvolution analyses.”*
- We have added text to the “Discussion” section that includes what work needs to follow up the study, in particular, text on the usefulness of single-cell RNAseq to investigate heterogeneity *“A challenge remains to better characterise immune evasion seen in the majority of OAC patients⁶⁶. Future work to study samples pre and post-treatment to investigate the immune clusters, and spatial impact of cells within the TME using single-cell and spatial transcriptomics will provide more insights into the interplay between the tumour and the microenvironment during treatment”.*

REVIEWER #2

Naeini et al performed combined genomic (WGS) and transcriptomic profiling on a cohort of treatment naïve esophageal adenocarcinomas in which post therapy (as part of the DOCTOR trial) patient outcome was known in order to identify potential determinants of overall survival. They identified several genomic features that were associated with overall survival; including APOBEC and RS3-like rearrangement signature, and increased copy number based sub-clonal heterogeneity associated with poor survival/worse response to therapy. Using the transcriptomic data, they were also able to classify the EACs into 4 immune categories and showed that the immune based clusters were associated with survival. Overall, this was a well performed study and the known details of treatments each patient received along with the combined WGS and transcriptomics is a major strength.

RESPONSE: We appreciate the reviewer for considering our study and highlighting the strengths of WGS and detailed treatment and clinical information.

One potential weakness of the study is that several similar recent manuscripts have been published that may take some away from the novelty/significance.

RESPONSE: This is a similar comment to reviewer number 1, we have highlighted the novelty of our studying when addressing the first comment to reviewer number 1 – see above.

1: For the cases with a high RS3-like signature, the authors state no alterations in BRCA1 or BRCA2 were identified. Were any alterations in other HR related genes identified (PALB2, RAD50,...)? If so, that could potentially explain the similarities while still being somewhat different.

RESPONSE: We thank the reviewer for raising this point. We have refined our analysis of HR deficiency to better determine which samples may be associated with HR deficiency (HRD). Originally, we used HRD score¹⁰ to determine samples with HRD. To improve this analysis we have now run HRDetect¹¹. Using these tools, we have assumed a sample is HR deficient based on published thresholds of HRD score > 42 and HRDetect score > 0.7. In the oesophageal cohort, 65 samples contained an HRD sum score > 42, nine samples contained an HRDetect > 0.7. Considering both tools, nine samples (**Supplementary Figure 2d and Supplementary Table 2**) contained HRD score > 42 AND HRDetect score > 0.7, and therefore we consider these as HR deficient. As mentioned in the results, we did not see a significant correlation between the HRD score and RS3-like and RS5-like signatures. We also investigated the correlation between HRD scores and HRDetect scores (Pearson correlation, $r=0.4$; $p=0.00012$) (**Supplementary Figure 2d**).

For the nine samples with HRD sum score > 42 AND HRDetect > 0.7, we have now investigated germline alterations in 27 HR-related genes and somatic changes in *BRCA1* or *BRCA2* (**Supplementary Table 3**) and did not find any candidate pathogenic or likely pathogenic germline variants or somatic mutations to explain the HRD in the 9 samples. We also investigated the seven samples with the highest R3-like signature (> 20% percent) and again we could not identify germline variants linked to HRD. We did identify germline pathogenic or likely pathogenic variants in the *ATM* (n=3 patients) and *NBN* (n=1 patient) genes. Furthermore, we investigated methylation levels in the *BRCA1* gene region. Here we found no tumour with hypermethylation in the promoter region of *BRCA1* gene. Moreover, we compared methylation levels in eight HRD and 61 non-HRD samples with available methylation profiles and did not find any methylation differences between the two groups (**Rebuttal Figure 2**).

Rebuttal Figure 2: Boxplots of *BRCA1* methylation values in samples with HRD (n=8, red) and without HRD (n=61, blue).

We conclude that the high TMB and a large number of SVs within the cohort with a low prevalence of HRD and lack of *BRCA1/2* mutations suggest that DNA repair may be perturbed but it is not classic BRCA-driven HRD.

CHANGES TO MANUSCRIPT:

- We have included the HRDetect analysis in the “Methods” section “We estimated levels of homologous recombination deficiency (HRD) using two distinct approaches scarHRD R package¹² (<https://github.com/sztup/scarHRD>) and HRDetect¹¹ (<https://github.com/eyzhao/hrdetect-pipeline>). The scarHRD package uses next-generation sequencing data including WGS and WES to calculate levels of telomere allelic imbalance (tAI), loss of heterozygosity (LOH) and the number of large-scale transitions (LST). scarHRD computes the unweighted numeric sum of LOH, tAI, and LST reported as HRD-sum¹², and represents HRD score in this study. The HRDetect package is a WGS and WES-based classifier predicting *BRCA1* and *BRCA2* deficiencies. The package uses SNV signatures, rearrangement signatures and HRD index including tAI, LOH and LST scores. The package computes an HRDetect score, a probability value quantifying the degree of *BRCA1/BRCA2* defectiveness. In this study, we classified samples as HRD with an HRD score >42 and HRDetect score >0.7.”
- We have included HRD score and HRDetect scores for each sample in **Supplementary Table 2**.
- We included a new **Supplementary Table 3** that contains all somatic SNVs and INDELS within coding regions in all patients. This table is in a MAF file format This table includes the somatic mutations in the 27 HR-related genes.

- We added text in the “Results” section to include these new analyses: *“Using HRDetect²⁴ and HRD scores²⁵ we identified nine samples that were HRD (**Supplementary Figure 2d**). Interestingly, the HRD samples did not harbour germline variants classified as “likely-pathogenic” or “pathogenic” (in ClinVar) in HR-related genes²⁶. One of the HRD samples (OESO_0047) did contain a somatic BRCA1 missense mutation, however, an HR proficient tumour (OESO_0118) contained a frameshift BRCA1 mutation (predicted pathogenic) (See **Supplementary Table 3** for all somatic coding mutations in the cohort). We also investigated BRCA1 methylation levels in the cohort and did not identify any tumour with hyper-methylation of the promoter region of the BRCA1 gene. We also compared BRCA1 methylation levels in eight HRD samples and 61 non-HRD samples with available methylation profiles and did not identify any association between HRD and methylation changes in the BRCA1 gene region. Taken together, these findings suggest the high TMB and a large number of SVs may be linked to perturbed DNA repair that is not classic BRCA-driven HRD.*
- We added text in the “Discussion” section. *“However, pathogenic somatic or germline mutations in HR-related genes were lacking and we found few samples considered as HRD according to HRDetect/HRDscores, indicating that PARP inhibitors may be ineffective and that an alternate DNA repair may be perturbed.”*

2: For the survival analysis comparing APOBEC, while statistically significant, a HR of 1.01 to 1.1 is quite low and doesn’t seem clinically relevant? Given how close it is to one, confirmation in an independent data set, such as the TCGA used later in the manuscript, would be valuable. The same is true for the RS-3 and RS-5 like signatures.

RESPONSE: We performed a new mutational signature analysis on TCGA using the same processes as for the WGS data (described in the Methods section). A caveat is that TCGA is whole-exome sequencing while our cohort includes whole-genome sequencing data. According to the COSMIC database, APOBEC-associated mutational signatures are signatures 2 and 13. Within the TCGA cohort, we only identified signature 13 with a cosine similarity 0.89 (new **Supplementary Figure 4a**). Using signature 13 we performed a log-rank survival test comparing APOBEC present and absent samples using the same threshold as the WGS cohort (APOBEC present ≥ 15). We identified a similar proportion of patients with APOBEC present and absent in TCGA compared to the WGS cohort. Although not significant, we noticed a similar trend for overall survival for APOBEC present and absent patients in the TCGA cohort (new **Supplementary Figure 4b**). A potential reason for the lack of significance in the survival association is the difference in sequence platform (WGS and exome) and that the samples within TCGA were collected several years prior to our WGS cohort and patients within TCGA generally have lower overall survival than the WGS cohort (new **Supplementary Figure 4c**).

In terms of RS3 and RS5-like signatures, these structural rearrangement signatures require whole-genome sequencing to identify breakpoint information and cannot be identified in whole-exome sequencing data. As we don’t have access to other WGS cohorts, we are not able to perform rearrangement signature analysis on a separate cohort.

CHANGES TO MANUSCRIPT:

- We have added **Supplementary Figure 4** describing the work to validate the APOBEC mutational signature and survival in the TCGA cohort. This includes the contribution of the APOBEC 13 signature in samples, a Kaplan-Meier plot and an overview of the overall patient survival in TCGA and our WGS cohort.
- We added text to the “Result” section: “*The APOBEC signature was also present in TCGA samples (Supplementary Figure 4a). The pattern of OS in TCGA was similar to our cohort, with absent APOBEC signature tended to have better OS, although this was not significant (Supplementary Figure 4b). The differences between our cohort and TCGA may be due to the differences in OS between the WGS and TCGA cohorts (Supplementary Figure 4c)*”.

3: It looks like high ploidy also correlated with stage in the 'non complex event' group. Could ploidy and stage be related? (ie could more advanced cancers have time to develop higher ploidy or alternatively could high ploidy drive progression?)

RESPONSE: We thank the reviewer for the interesting question. We performed a new analysis to address this comment and found a correlation between ploidy and tumour stage in samples in the “non-complex event” group (Pearson $r=0.41$, $p=0.014$) (new **Supplementary Figure 6**).

We additionally explored the relationship through the following analysis. In the “non-complex” group, there were $n=24$ samples with a high ploidy (ASCAT ploidy >2.7), $n=13$ samples with a high stage (stage III and IV), and $n=11$ samples were both high ploidy and a high stage. We computed the following proportions:

$$P1 = \frac{\text{high ploidy and high stage}}{\text{high stage}} = \frac{11}{13} = 84.6\%$$

$$P2 = \frac{\text{high ploidy and high stage}}{\text{high ploidy}} = \frac{11}{24} = 45.8\%$$

Proportion 1 (P1) is almost double compared to Proportion 2 (P2). Using binomial test ($x=11$, number of trials=13 and null hypothesis= 0.458) confirms the significant difference between the two proportions (p -value = 0.009082). This means that high-stage samples (stage III and IV) within the “non-complex group” are enriched with high ploidy tumours, while the same is not necessarily true for high ploidy samples as high ploidy samples are not necessarily high-stage or advanced tumours. These findings suggest that high ploidy is a late event in tumorigenesis occurring in stage II/IV tumours. Recently, it was proposed that polyploidy in oesophageal adenocarcinoma originates from mitotic slippage caused by defective chromosome attachments¹³. Our findings suggest this may be a mutational mechanism that is a feature of late-stage oesophageal cancer.

CHANGES TO MANUSCRIPT:

- We added a new **Supplementary Figure 6** showing the association of stage and ploidy.
- We have amended the text in the “Results” section to include the correlation between ploidy and stage “*In tumours that did not harbour complex events, high ploidy was associated with worse OS (log-rank, $p=0.015$, $q=0.03$) (Figure 3b) and late tumour stage (Pearson $r=0.41$, $p=0.014$) (Supplementary Figure 6a and b)*”.
- We have amended and added text to the “Discussion” section: “*Moreover, patients with tumours that did not contain complex genomes, but were high tumour ploidy had worse survival.*”

Recently, it was proposed that polyploidy in oesophageal adenocarcinoma originates from mitotic slippage caused by defective chromosome attachments³², our findings suggest this may be a mutational mechanism that is a feature of late-stage oesophageal cancer and warrants further investigation”.

4: The fact cluster 3 had a lower Sig 17 is interesting. Could this give any insight into the cause of this signature? (ie does cluster 3 have something uniquely missing compared to the other clusters that could induce Sig 17 like mutations?)

RESPONSE: We agree that lower Signature 17 in Cluster 3 is interesting. Notably, we did not find any mutations related to Signature 17. However, we showed that signature 17 is correlated with a higher SNV burden (**Figure 2b**), with the range of mutation burden in Cluster 3 (range 3.1 to 6.2 mutations/Mb) being lower than the average mutation burden across the cohort (8.35 mutations/Mb). Pathway analysis suggested the DNA repair pathway is lower in Cluster 3. Therefore, the lower levels of signature 17 within Cluster 3 could potentially be explained by the lower levels of error-prone DNA repair pathways, or by an environment with less oxidative damage.

Furthermore, we and others found a correlation between SNV mutation burden and neoantigen levels¹⁴ (**Supplementary Figure 1a**). Previous studies found a higher number of neoantigens in immune hot patients and a possible cause for activating T-cells. In support of this, we found a lower Signature 17 and TMB in Cluster 3 which is the immune moderate group with lower T-cells infiltrate.

CHANGES TO MANUSCRIPT:

- We added a text to the “Result” section and discussed DNA repair pathway in Cluster 3 *“Cluster 3 (immune moderate) uniquely demonstrated lower levels DNA repair pathway.”*
- We have amended and added text to the “Discussion” section: *“Cluster 3 tumours were marked by higher p53 pathway level, lower DNA repair pathways, lower mutation burden and a lower proportion of mutation signature 17. The aetiology of signature 17 remains unclear, but in our cohort, it may be caused by oxidative damage²⁰ during digestion or chronic reflux as opposed to treatments with 5-FU¹⁶, as the OAC tumours in our cohort were collected prior to any treatment. Therefore, the lower levels of signature 17 within Cluster 3 could potentially be explained by the lower levels of error-prone DNA repair pathways, or by an environment with less oxidative damage. In support of previous reports⁵⁴, we found a correlation between neoantigens and SNV mutation burden. Although we did not identify a higher neoantigen burden in Cluster 1 “immune hot”⁵⁵, we did see a lower SNV mutation burden in Cluster 3 which was linked with moderate infiltrations of lymphocytes.”*

5: The authors should be a little careful between cause and effect vs associations in the discussion. The authors appropriately use qualifiers, but some statements may still need to be toned down. For example, line 287+, with the given data it is difficult to conclude that CF treatment improved survival because the RS-5 like signature is present. Just as likely, an alternative would be a biological process that is creating RS-5 causes less aggressive cancers or responds to CF treatment.

RESPONSE: Many thanks for this comment, we have modified parts of the discussion accordingly.

CHANGES TO MANUSCRIPT:

- we have amended the “Discussion” throughout to tone down some of the cause vs effect comments and future work to confirm our findings, including:
- For RS5-like signature the “Discussion” has been amended: *“Interestingly, the RS5-like signature was higher in patients with better overall survival, however, we are unable to conclude if this is due to tumours with an RS5-like signature responding to DNA damaging agents (cisplatin and 5-Fluorouracil) or whether other factors such as the possibility of RS5-like tumours being less aggressive may be linked with the better response. Therefore, further investigation would need to be conducted to confirm the patient survival effect of the RS5-like signature.”*
- For the methylation analysis, we have included in the “Discussion”: *“Methylation profiling did not suggest an association between methylation changes and complex events at EGFR, MTMR9, GATA4, GATA6, and SMAD4 genes regions. However, aberrant methylation has been previously linked to tumours with more SV events and tumours with focal amplifications⁴⁰. Future studies focusing on ATAC-seq or direct DNA long read sequencing to resolve complex events and simultaneously profile methylation may provide insights into the mechanisms surrounding the complex genomic events”*
- A new section in the discussion highlights current and future work using preclinical models, such as organoids, to confirm findings.

6: A major issue within the manuscript is the lack of comparison to a few other recently published studies (Jammula Gastro 2020, Goedegebuure OncoImmunology 2021, de Klerk Mol Onco 2021, Derks Annals of Onco 2020) who performed (at least somewhat) similar experiments, although maybe less robustly. Nevertheless, it does somewhat call into question the novelty of the current study and as such some of the discussion should be devoted to compare the authors results to the prior work to stress the novel findings and show how the studies may complement each other.

RESPONSE: Many thanks for highlighting these papers, we have reviewed them in turn and provided the following detailed response. In summary, we believe our study not only provides critical validation of previous work, but a much deeper insight into the genome and tumour microenvironment of pre-treated tumours with neoadjuvant therapy that are mostly novel and complementary to previous findings.

RESPONSE: The study from De Klerk et al used targeted panel sequencing to identify genomic alterations in oesophageal adenocarcinoma (n=75) and oesophageal squamous cell carcinoma (n=16) patients treated with neoadjuvant chemoradiotherapy. De Klerk et al. found that *CSMD1* deletion (8%) and *ETV4* amplification (5%) were associated with a favourable histopathological response, whereas *SMURF1* amplification (5%) and *SMARCA4* mutation (7%) were associated with an unfavourable histopathological response. In addition, *KRAS* (15%) and *GATA4* (7%) amplification were associated with shorter OS.

In our cohort, we could not confirm these findings as did not see an association between these genes and histopathology response (**Rebuttal Figure 3a**), and these genes were not associated with overall survival (**Rebuttal Figure 3b and c**) (*KRAS* p-value=0.45, *GATA4* p-value=0.93). The reason why some of

the markers reported by De Klerk et al. do not validate in our study could be due to the inclusion of different patients or methodological differences such as the WGS approach we have used. The WGS we have used did allow us to expand our analyses to mutational signatures rearrangement signatures, kataegic loci and complex events – all of which were not possible with the targeted sequencing from De Klerk et al.

CHANGES TO MANUSCRIPT: We have amended the “Result” to include a comparison to the findings from De Klerk et al: *“The presence of SMGs was not associated with histopathological response or overall survival. Furthermore, we did not validate a previous report²¹, of histopathology response being associated to somatic copy number changes in CSMD1, ETV4, SMURF1 and mutations in SMARC4 and amplification of KRAS and GATA4 being associated with overall survival.”*

Rebuttal Figure 3: a) Oncoplot of genes reported by Klerk et al. and histopathology response. b) Kaplan Meier plot of b) KRAS amplification and overall survival c) GATA4 amplification and overall survival. Copy Number Deletion (CN \leq 1); Copy Number Highgain (CN \geq 6).

RESPONSE: The study from Jammula et al identified subtypes of Barrett's esophagus (BE) and oesophageal adenocarcinoma using DNA methylation profiles and they further discussed the cell type composition of each subtype using bulk RNA-seq. Using the methylation profiles, they identified four distinct subtypes, the first (subtype 1) was comprised of tumours with a CIMP-phenotype. We have previously characterized CIMP-like OAC samples in our earlier work (DOI: 10.1093/carcin/bgw018). The

second subtype reported by Jammula et al (subtype 2) mostly consisted of BE samples, therefore we can't draw any comparisons to this subtype. The third methylation subtype reported by Jammula et al (subtype 3) had a strong enrichment of both innate and adaptive immune cell types, so this may be similar to our "immune hot" cluster identified by RNAseq. The fourth subtype reported by Jammula et al (subtype 4) was reported to have general hypomethylation with more SV events and tumours with focal amplifications, therefore this subtype aligns well with the complex genome group we have identified, where we see methylation changes associated with complex rearrangements.

Therefore, we conclude that the RNA clusters that we describe in this paper are not directly linked to the methylation groups. This is not unexpected. In the Jammula et al. study, for the methylation profiling, they used samples with tumour purity >70% which is very suitable for genomic analyses and identifying subgroups of tumours samples based on their methylation status. However, this approach would limit the study of the tumour microenvironment, as DNA from the immune cells will be less abundant in the sample. In contrast, when we use RNAseq to identify the cluster, we find an association with immune cell abundance as the RNASeq has a higher dynamic range and the variable gene expression allows deconvolution of the cells in the TME.

RESPONSE: The study from Derks et al. investigated the tumour microenvironment of gastroesophageal adenocarcinomas subtypes including MSI, EBV, CIN and Genome stable (GS) using TCGA exome sequence and RNAseq data. The majority of this data was from gastric tumours, therefore the findings are not directly comparable with our findings. However, this paper did inspire us to conduct a pathway analysis of genes associated with each cluster. In agreement with Derks et al, we did find pathways such as interferon-gamma response and IL2/STAT5 signaling in the immune hot group (**Figure 5g**). Furthermore, Derks et al identified IL6/JAK/STAT, KRAS, and TNFa signaling via NFkB as enriched in the immune hot group, we confirm this enrichment, but also found enrichment in the immunosuppressed tumours in Cluster 2 (**Figure 5g-h**).

CHANGES TO MANUSCRIPT: To compare the pathways detected by Derks et al., we have performed pathway analysis which resulted in the following changes to the manuscript:

- Added text to the "Methods" section to detail the pathway analysis: *"Pathway enrichment analysis was performed using pre-ranked Gene Set Enrichment Analysis (GSEA) and Hallmark "H" gene signatures from the Molecular Signatures Database (MSigDB, <http://www.gsea-msigdb.org/gsea/msigdb/collections.jsp>). Here the GSEA input was prepared by ranking the genes using log-fold changes from differential expression analysis. Pathways with adjusted p-value <0.05 were included in analyses."*
- We have added two new panels to **Figure 5g-h** and **Supplementary figure 18a-b** showing the enriched pathways in each of the four immune clusters.
- We added text to the "Result" section describing the results from the enrichment analysis: *"We performed gene set enrichment analysis (GSEA) using "Hallmark" gene signatures to identify signalling pathways in each cluster. Cluster 1 (immune hot, better survival) was uniquely enriched with Interferon alpha/gamma responses, IL2/STAT5 pathway, apoptosis, hypoxia and p53 pathway, and showed lower levels of cell cycle and progression pathways such as E2F targets and G2M checkpoint (**Figure 5g**). Cluster 2 (immunosuppressed, worse survival) was uniquely enriched with cell cycle and progression pathways such as E2F targets, G2M checkpoint,*

and represented a lower level of p53 pathway (**Figure 5h**). Both Clusters 1 and 2 showed activation of KRAS signalling pathway. Cluster 3 (immune moderate) uniquely demonstrated lower levels DNA repair pathway. Cluster 3 also showed a higher level of p53 pathway and lower levels of immune responses as well as apoptosis (**Supplementary Figure 18a**). Cluster 4 (immune cold) was enriched with metabolism pathways such fatty acid metabolism and was high in protein secretion (**Supplementary Figure 18b**)."

- We have included text in the Discussion: "Similar to previous reports in gastroesophageal adenocarcinomas⁴⁹ the "immune hot" tumours in Cluster 1 were enriched with cytokine and immune-related pathways such as interferon alpha/gamma, EMT as well as P53 pathway. While Cluster 1 tumours showed lower levels of cell cycle and progression pathways such as E2F and G2M checkpoints."

RESPONSE: The recent study from Goedegebuure et al. investigated pre-treated tumours of oesophageal adenocarcinoma using multiplex immunohistochemistry, flow cytometry and NanoString mRNA expression. They used this data to discuss the influence of the tumour microenvironment in response to neoadjuvant chemoradiation therapy. Compared to the studies above, Goedegebuure et al. is more relevant to our study. In Figure 3a of Goedegebuure et al., shows that complete responders (CR) have a higher cytotoxic T cell relative to total leukocytes, in Figure 3c they also report genes that are differentially expressed in CR compared to non-responders. In our data, we see that many enriched genes, including *CD8A*, *GNLY*, *CCL5*, *CXCL9* and *CXCL10*, which were associated with a CR in Goedegebuure et al. were identified in our Cluster 1 (immune hot). Therefore, we assume the complete responders (CR) with high cytotoxic T-cells in Goedegebuure et al. represent Cluster 1 (immune hot) in our study. In addition, the gene most associated with non-response in Goedegebuure et al. was *SPP1*, we also found enrichment of *SPP1* in Cluster 2 (immunosuppressed) tumours. Therefore, our study is consistent with the Goedegebuure et al. study, but our comprehensive analysis (WGS and RNAseq) allows us to better stratify the patients.

CHANGES TO MANUSCRIPT: We have added new text to the "Discussion" section that highlights the Goedegebuure et al. study. "Previous studies reported *CCL5*, *NKG7*, *CXCL9*, and *CXCL10* as biomarkers for better survival in oesophageal squamous carcinoma and other cancer types^{48,49,50}, and high expression of *CD8A*, *GNLY*, *CCL5*, *CXCL9* and *CXCL10* in pre-treated OAC tumours with high T-cell infiltrates was linked to complete response to neoadjuvant chemoradiotherapy³⁰." AND "*SPP1* (Osteopontin) is secreted by tumour cells and myeloid cells and is associated with aggressiveness and immune suppression in the TME⁵² and has been reported as enriched in pre-treated OAC tumours which did not respond to neoadjuvant chemoradiotherapy³⁰."

6b: In Jammula et al, through bulk RNA-seq immune analysis they also identify an 'immune hot' molecular subtype. However, in this study the 'immune hot' subtype has the worst prognosis. Based on what Jammula et al reported, is there any obvious differences between the subtypes that might give clues on the importance of individual cell types on survival?

RESPONSE: As discussed in the previous comment, we think the subtypes represented in Jammula et al are different from the identified immune clusters in our study. Therefore, the survival findings in Jammula et al. are not comparable to the survival associations in our study. According to Jammula et al. Figure 5e (survival analysis), subtype 2 (which is linked with Barrett's oesophagus samples) has the best

overall survival with the most separated survival curve. The reported p-value in Figure 5e of Jammula et al. was p-value= 0.038, therefore if subtype 2 was removed the other subtypes may not be significantly associated with survival. In our study, we explain the features of each immune cluster and why they may be associated with survival.

6c: Similar to the neutrophil/T-cell ratio used in this study, In Goedegebuure et al found that a high CD8:CD163 ratio was associated with improved disease-free survival. While the neutrophil/T-cell ratio was high in clusters 2 and 4, are the CD8/CD163 ratios similar or different in the clusters (ie does it better correlate with treatment response/survival)?

RESPONSE: Goedegebuure et al found high CD8:CD163 ratio was associated with improved disease-free survival. We found that Cluster 2 (immunosuppressed, worse survival) has a higher neutrophil/T-cell ratio and Cluster 1 (immune hot, better survival) has a lower neutrophil/T-cell ratio. We also found that there is a positive correlation between neutrophils, macrophages and monocytes (high CD163). Accordingly, both studies are in agreement and complement each other.

CHANGES TO MANUSCRIPT: We have added new text to the “Result” section that mentions the consistency of Goedegebuure et al. findings with our findings. *“This supports a previous finding of an association of high CD8:CD163 ratio in OAC, comparable with Cluster 1 in our study, with an improved disease-free survival³⁰”*

7: This reviewer appreciated the well described methods section. It served to increase the confidence in the results.

RESPONSE: We thank the reviewer for appreciating the well-described methods section.

Minor:

1: Line 72-77, the authors may want to adjust the sentence to note that while signature 17 is associated with 5-FU treatment, in OAC something else is likely causing the signature. As the authors are aware, signature 17 is also the dominate signature in Barrett's oesophagus and OAC not exposed to 5-FU (as in this manuscript).

CHANGES TO MANUSCRIPT: We have revised the sentence to clarify the presence of signature 17, it now reads *“Signature 17 is a dominant signature in OAC¹⁵, it can be an early event as was detected in Barrett’s oesophagus^{16,17}, and can also occur later as has been associated with 5-fluorouracil (5-FU) treatment¹⁸”*.

2: While being well defined in the methods, a quick ½ sentence on the definition of complex events in the results (line 155) would aid in reading if there is space.

CHANGES TO MANUSCRIPT: We have amended a sentence in the “Results” section to define the complex events. *“We identified tumours with complex events including chromothripsis, BFB or other clustered events,”*

3: For Figure S4, 'del' is defined as homozygous deletion correct?

CHANGES TO MANUSCRIPT: The “del” label refers to homozygous deletion (Copy number 0) and heterozygous deletions (copy number 1). We added a sentence to **Supplementary Figure 5** (previously 4) legend to clarify the confusion: “Copy number (CN) is coloured by HighGain (CN ≥ 6, red), Del (CN0 and CN1, blue) and cnLOH is copy number neutral LOH (CN2, yellow) are shown”

4: Line 321, The manuscript shows enriched myeloid cells in cluster 2, but unsure if it shows MDSCs. It could have been missed, but if not, please be more specific on the cell types.

CHANGES TO MANUSCRIPT: We have amended the sentence to remove the word “*suppressor*” as our intention was to refer to myeloid cells.

5: Line 522-532: Paragraph seems duplicated from the one above.

CHANGES TO MANUSCRIPT: We thank the reviewer for this comment. We have removed the duplicated paragraph in the Methods section of the revised manuscript.

References

1. Kato, K., *et al.* LBA8_PR Pembrolizumab plus chemotherapy versus chemotherapy as first-line therapy in patients with advanced esophageal cancer: The phase 3 KEYNOTE-590 study. *Annals of Oncology* **31**, S1192-S1193 (2020).
2. Shah, M.A., *et al.* Efficacy and safety of pembrolizumab for heavily pretreated patients with advanced, metastatic adenocarcinoma or squamous cell carcinoma of the esophagus: the phase 2 KEYNOTE-180 study. *JAMA oncology* **5**, 546-550 (2019).
3. Kojima, T., *et al.* Pembrolizumab versus chemotherapy as second-line therapy for advanced esophageal cancer: Phase III KEYNOTE-181 study. (American Society of Clinical Oncology, 2019).
4. Chakravarthy, A., *et al.* Pan-cancer deconvolution of tumour composition using DNA methylation. *Nature Communications* **9**, 3220 (2018).
5. Buenostro, J.D., Wu, B., Chang, H.Y. & Greenleaf, W.J. ATAC-seq: a method for assaying chromatin accessibility genome-wide. *Current protocols in molecular biology* **109**, 21.29. 21-21.29. 29 (2015).
6. Goryshin, I.Y. & Reznikoff, W.S. Tn5 in vitro transposition. *Journal of Biological Chemistry* **273**, 7367-7374 (1998).
7. Adey, A. & Shendure, J. Ultra-low-input, tagmentation-based whole-genome bisulfite sequencing. *Genome research* **22**, 1139-1143 (2012).
8. Wu, H., *et al.* Single-cell RNA sequencing reveals diverse intratumoral heterogeneities and gene signatures of two types of esophageal cancers. *Cancer letters* **438**, 133-143 (2018).
9. Hao, Y., *et al.* Integrated analysis of multimodal single-cell data. *Cell* (2021).
10. Telli, M.L., *et al.* Homologous recombination deficiency (HRD) score predicts response to platinum-containing neoadjuvant chemotherapy in patients with triple-negative breast cancer. *Clinical cancer research* **22**, 3764-3773 (2016).
11. Davies, H., *et al.* HRDetect is a predictor of BRCA1 and BRCA2 deficiency based on mutational signatures. *Nature medicine* **23**, 517 (2017).

12. Sztupinski, Z., *et al.* Migrating the SNP array-based homologous recombination deficiency measures to next generation sequencing data of breast cancer. *npj Breast Cancer* **4**, 16 (2018).
13. Scott, S.J., *et al.* Evidence that polyploidy in esophageal adenocarcinoma originates from mitotic slippage caused by defective chromosome attachments. *Cell Death Differ* **28**, 2179-2193 (2021).
14. Gupta, R.G., Li, F., Roszik, J. & Lizée, G. Exploiting tumor neoantigens to target cancer evolution: current challenges and promising therapeutic approaches. *Cancer Discov* **11**, 1024-1039 (2021).
15. Secrier, M., *et al.* Mutational signatures in esophageal adenocarcinoma define etiologically distinct subgroups with therapeutic relevance. *Nature genetics* **48**, 1131-1141 (2016).
16. Newell, F., *et al.* Complex structural rearrangements are present in high-grade dysplastic Barrett's oesophagus samples. *BMC Med Genomics* **12**, 31 (2019).
17. Stachler, M.D., *et al.* Paired exome analysis of Barrett's esophagus and adenocarcinoma. *Nature Genetics* **47**, 1047-1055 (2015).
18. Christensen, S., *et al.* 5-Fluorouracil treatment induces characteristic T>G mutations in human cancer. *Nature Communications* **10**, 4571 (2019).

REVIEWERS' COMMENTS

Reviewer #1 (Remarks to the Author):

I have no additional comments except that my original concerns remain.

Reviewer #2 (Remarks to the Author):

The authors have made extensive revisions to the manuscript and included additional data. All of this reviewers comments/questions were addressed. One minor, optional suggestion would be to comment that the HRD samples did not harbor germline or somatic variants in HR related genes classified as likely pathogenic or pathogenic as opposed to focusing on germline ("Interestingly, the HRD samples did not harbour germline variants classified as "likely-pathogenic" or "pathogenic" (in ClinVar) in HR-related genes"). As I would suspect most readers would guess any HRD related variant would probably arise as a somatic event in OAC. This is clarified later in the manuscript, but still might be worth mentioning up front. It was interesting that the two HRD algorithms were so different in calling HRD deficiency. I do have a tendency to agree with the more conservative one and I wonder if HRDscore overcalls in samples with a general high degree of CNVs. Providing the SNV/Indel table with all exonic mutations is a nice addition for any reader who wants to explore something specific.

Reviewer #3 (Remarks to the Author):

In my opinion, the most significant findings in this manuscript (NCOMMS-21-33901A) include: 1) the relatively complete genomics and epigenomics profiling of a fairly sizable cohort of EAC with complete clinical data; 2) very solid computational and statistical analyses on the genomics data and the associations with clinical outcomes/parameters; 3) the 4 clusters identified by RNA-seq, which can be linked to immune responses and survival.

Regarding the Reviewer#1's comment: "3. The molecular analysis would be strengthened by consideration of epigenetic changes through use of ATAC-seq. Furthermore, single-cell ATAC-seq/RNA-seq would be useful to highlight tumor heterogeneity as a complex issue.", I think the authors have addressed it fairly adequately.

The authors replied that they addressed epigenetics using methylation analyses. While I opine that the methylation data itself actually does not really add much to either the significance or novelty for this current paper, I doubt whether new ATAC-seq is that necessary, let alone that it is impossible to do at this stage. Regarding the comment on single-cell ATAC-seq/RNA-seq, I do not think that is the main point of this paper. They might be able to contribute to the present work in way of validation/confirmation, but I do not envision that these data can substantially change/improve the work.

Responses to reviewers

We thank the reviewers for their comments. In this document we provide a response to the comment from reviewer 2.

REVIEWER #2

The authors have made extensive revisions to the manuscript and included additional data. All of this reviewers comments/questions were addressed. One minor, optional suggestion would be to comment that the HRD samples did not harbor germline or somatic variants in HR related genes classified as likely pathogenic or pathogenic as opposed to focusing on germline ("Interestingly, the HRD samples did not harbour germline variants classified as "likely-pathogenic" or "pathogenic" (in ClinVar) in HR-related genes"). As I would suspect most readers would guess any HRD related variant would probably arise as a somatic event in OAC. This is clarified later in the manuscript, but still might be worth mentioning up front.

RESPONSE: We have amended the text to refer to germline and somatic variants in these cases. The text now reads

*"Interestingly, the HRD samples did not harbour germline variants classified as likely-pathogenic or pathogenic (in ClinVar) in HR-related genes²⁵. One of the HRD samples (OESO_0047) did contain a somatic BRCA1 missense change, however, an HR proficient tumour (OESO_0118) also contained a frameshift BRCA1 mutation (predicted pathogenic) and 3 other HR proficient samples contained somatic BRCA1 or BRCA2 missense changes (See **Supplementary Data 3** for all somatic coding mutations in the cohort)."*

It was interesting that the two HRD algorithms were so different in calling HRD deficiency. I do have a tendency to agree with the more conservative one and I wonder if HRDscore overcalls in samples with a general high degree of CNVs. Providing the SNV/Indel table with all exonic mutations is a nice addition for any reader who wants to explore something specific.

RESPONSE: This is interesting, as if HRDscore is affected by a high degree of CNV, it is likely the approach will be impacted in OAC where a high degree of somatic CNV exists. Furthermore, we agree with the reviewer that the HRDetect approach, which is more conservative, appears to be a better predictor of HRD, at least in breast cancer. In support of this, recently some of our author team were part of a paper submitted to Medrxiv (<https://doi.org/10.1101/2023.02.21.23286199>), which used WGS of breast cancers and shows that the more conservative approach had a better alignment with cases harboring BRCA1/2 mutations (Figure 5 of that paper).